# Learning Diverse and Discriminative Representations via the Principle of Maximal Coding Rate Reduction

**Yaodong Yu**[†]
UC Berkeley
yyu@eecs.berkeley.edu

**Kwan Ho Ryan Chan**[†]
UC Berkeley
ryanchankh@berkeley.edu

**Chong You**
UC Berkeley
cyou@berkeley.edu

**Chaobing Song**
Tsinghua University
songcb16@mails.tsinghua.edu.cn

**Yi Ma**
UC Berkeley
yima@eecs.berkeley.edu

## Abstract

To learn intrinsic low-dimensional structures from high-dimensional data that most discriminate between classes, we propose the principle of *Maximal Coding Rate Reduction* (MCR$^2$), an information-theoretic measure that maximizes the coding rate difference between the whole dataset and the sum of each individual class. We clarify its relationships with most existing frameworks such as cross-entropy, information bottleneck, information gain, contractive and contrastive learning, and provide theoretical guarantees for learning diverse and discriminative features. The coding rate can be accurately computed from finite samples of degenerate subspace-like distributions and can learn intrinsic representations in supervised, self-supervised, and unsupervised settings in a unified manner. Empirically, the representations learned using this principle alone are significantly more robust to label corruptions in classification than those using cross-entropy, and can lead to state-of-the-art results in clustering mixed data from self-learned invariant features.

## 1 Context and Motivation

Given a random vector $\boldsymbol{x} \in \mathbb{R}^D$ which is drawn from a mixture of, say $k$, distributions $\mathcal{D} = \{\mathcal{D}_j\}_{j=1}^k$, one of the most fundamental problems in machine learning is how to effectively and efficiently *learn the distribution* from a finite set of i.i.d samples, say $\boldsymbol{X} = [\boldsymbol{x}_1, \boldsymbol{x}_2, \ldots, \boldsymbol{x}_m] \in \mathbb{R}^{D \times m}$. To this end, we *seek a good representation* through a continuous mapping, $f(\boldsymbol{x}, \theta) : \mathbb{R}^D \to \mathbb{R}^d$, that captures intrinsic structures of $\boldsymbol{x}$ and best facilitates subsequent tasks such as classification or clustering.

**Supervised learning of discriminative representations.** To ease the task of learning $\mathcal{D}$, in the popular supervised setting, a true class label, represented as a one-hot vector $\boldsymbol{y}_i \in \mathbb{R}^k$, is given for each sample $\boldsymbol{x}_i$. Extensive studies have shown that for many practical datasets (images, audios, and natural languages, etc.), the mapping from the data $\boldsymbol{x}$ to its class label $\boldsymbol{y}$ can be effectively modeled by training a deep network [GBC16], here denoted as $f(\boldsymbol{x}, \theta) : \boldsymbol{x} \mapsto \boldsymbol{y}$ with network parameters $\theta \in \Theta$. This is typically done by minimizing the *cross-entropy loss* over a training set $\{(\boldsymbol{x}_i, \boldsymbol{y}_i)\}_{i=1}^m$, through backpropagation over the network parameters $\theta$:

$$\min_{\theta \in \Theta} \ \text{CE}(\theta, \boldsymbol{x}, \boldsymbol{y}) \doteq -\mathbb{E}[\langle \boldsymbol{y}, \log[f(\boldsymbol{x}, \theta)]\rangle] \approx -\frac{1}{m} \sum_{i=1}^m \langle \boldsymbol{y}_i, \log[f(\boldsymbol{x}_i, \theta)]\rangle. \tag{1}$$

Despite its effectiveness and enormous popularity, there are two serious limitations with this approach: 1) It aims only to predict the labels $\boldsymbol{y}$ even if they might be mislabeled. Empirical studies show

---

[†]The first two authors contributed equally to this work.

that deep networks, used as a "black box," can even fit random labels [ZBH+17]. 2) With such an end-to-end data fitting, despite plenty of empirical efforts in trying to interpret the so-learned features [ZF14], it is not clear to what extent the intermediate features learned by the network capture the intrinsic structures of the data that make meaningful classification possible in the first place. The precise geometric and statistical properties of the learned features are also often obscured, which leads to the lack of interpretability and subsequent performance guarantees (e.g., generalizability, transferability, and robustness, etc.) in deep learning. Therefore, the goal of this paper is to address such limitations of current learning frameworks by reformulating the objective towards learning *explicitly meaningful* representations for the data $\boldsymbol{x}$.

**Minimal discriminative features via information bottleneck.** One popular approach to interpret the role of deep networks is to view outputs of intermediate layers of the network as selecting certain latent features $\boldsymbol{z} = f(\boldsymbol{x}, \theta) \in \mathbb{R}^d$ of the data that are discriminative among multiple classes. Learned representations $\boldsymbol{z}$ then facilitate the subsequent classification task for predicting the class label $\boldsymbol{y}$ by optimizing a classifier $g(\boldsymbol{z})$:

$$\boldsymbol{x} \xrightarrow{f(\boldsymbol{x}, \theta)} \boldsymbol{z}(\theta) \xrightarrow{g(\boldsymbol{z})} \boldsymbol{y}.$$

The *information bottleneck* (IB) formulation [TZ15] further hypothesizes that the role of the network is to learn $\boldsymbol{z}$ as the minimal sufficient statistics for predicting $\boldsymbol{y}$. Formally, it seeks to maximize the mutual information $I(\boldsymbol{z}, \boldsymbol{y})$ [CT06] between $\boldsymbol{z}$ and $\boldsymbol{y}$ while minimizing $I(\boldsymbol{x}, \boldsymbol{z})$ between $\boldsymbol{x}$ and $\boldsymbol{z}$:

$$\max_{\theta \in \Theta} \text{IB}(\boldsymbol{x}, \boldsymbol{y}, \boldsymbol{z}(\theta)) \doteq I(\boldsymbol{z}(\theta), \boldsymbol{y}) - \beta I(\boldsymbol{x}, \boldsymbol{z}(\theta)), \quad \beta > 0. \tag{2}$$

Given one can overcome some caveats associated with this framework [KTVK18], such as how to accurately evaluate mutual information with finitely samples of degenerate distributions, this framework has been successful in describing certain behaviors of deep networks. But by being task-dependent (depending on the label $\boldsymbol{y}$) and seeking a *minimal* set of most informative features for the task at hand (for predicting the label $\boldsymbol{y}$ only), the network sacrifices generalizability, robustness, or transferability, in case the labels can be corrupted or the learned features be tackled. To address this, our framework uses label $\boldsymbol{y}$ only as side information to assist learning *diverse* and *discriminative* representations, hence making learned features more robust to mislabeled data.

**Contractive learning of generative representations.** Complementary to the above supervised discriminative approach, *auto-encoding* [BH89, Kra91] is another popular *unsupervised* (label-free) framework used to learn good latent representations, which can be viewed as a nonlinear extension to the classical PCA [Jol02]. The idea is to learn a compact latent representation $\boldsymbol{z} \in \mathbb{R}^d$ that adequately regenerates the original data $\boldsymbol{x}$ to certain extent, through optimizing decoder or generator $g(\boldsymbol{z}, \eta)$:

$$\boldsymbol{x} \xrightarrow{f(\boldsymbol{x}, \theta)} \boldsymbol{z}(\theta) \xrightarrow{g(\boldsymbol{z}, \eta)} \widehat{\boldsymbol{x}}(\theta, \eta). \tag{3}$$

Typically, such representations are learned in an end-to-end fashion by imposing certain heuristics on geometric or statistical "compactness" of $\boldsymbol{z}$, such as its dimension, energy, or volume. For example, the *contractive* autoencoder [RVM+11] penalizes local volume expansion of learned features approximated by the Jacobian $\|\frac{\partial \boldsymbol{z}}{\partial \theta}\|$. Another key design factor of this approach is the choice of a proper, but often elusive, metric that can measure the desired *similarity* between $\boldsymbol{x}$ and the decoded $\widehat{\boldsymbol{x}}$, either between sample pairs $\boldsymbol{x}_i$ and $\widehat{\boldsymbol{x}}_i$ or between the two distributions $\mathcal{D}_{\boldsymbol{x}}$ and $\mathcal{D}_{\widehat{\boldsymbol{x}}}$. However, the distance between two distributions, say the KL divergence $\text{KL}(\mathcal{D}_{\boldsymbol{x}} || \mathcal{D}_{\widehat{\boldsymbol{x}}})$, is very difficult to evaluate when the data distributions are discrete and degenerate. In practice, it can only be approximated with the help of an additional disriminative network, known as GAN [GPAM+14, ACB17].

Representations learned through this framework can be arguably rich enough to regenerate the data to a certain extent. But depending on the choice of the regularizing heuristics on $\boldsymbol{z}$ and similarity metrics on $\boldsymbol{x}$ (or $\mathcal{D}_{\boldsymbol{x}}$), the objective is typically task-dependent and often grossly approximated [RVM+11, GPAM+14]. When the data contain complicated *multi-modal* structures, naive heuristics or inaccurate metrics may fail to capture all internal subclass structures or to explicitly discriminate among them for classification or clustering purposes. For example, one consequence of this is the phenomenon of *mode collapsing* in learning generative models for data that have mixed multi-modal structures [LPZM20]. To address this, we propose a principled measure (on $\boldsymbol{z}$) to learn representations that promotes multi-class discriminative property from data of mixed structures, which works in both supervised and unsupervised settings.

**This work: Learning diverse and discriminative representations.** Whether the given data $\boldsymbol{X}$ of a mixed distribution $\mathcal{D}$ can be effectively classified depends on how separable (or discriminative)

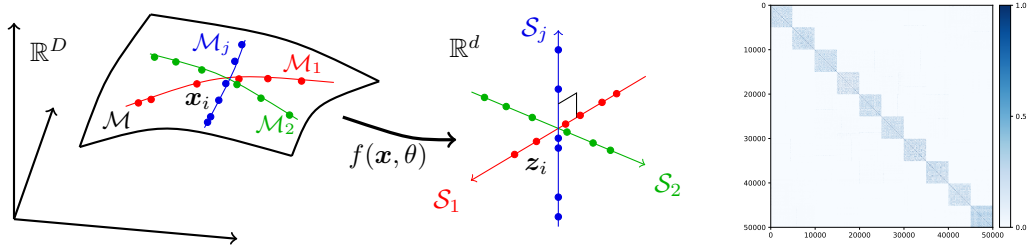

Figure 1: **Left and Middle:** The distribution $\mathcal{D}$ of high-dim data $\boldsymbol{x} \in \mathbb{R}^D$ is supported on a manifold $\mathcal{M}$ and its classes on low-dim submanifolds $\mathcal{M}_j$, we learn a map $f(\boldsymbol{x}, \theta)$ such that $\boldsymbol{z}_i = f(\boldsymbol{x}_i, \theta)$ are on a union of maximally uncorrelated subspaces $\{\mathcal{S}_j\}$. **Right:** Cosine similarity between learned features by our method for the CIFAR10 training dataset. Each class has 5,000 samples and their features span a subspace of over 10 dimensions (see Figure 3(c)).

the component distributions $\mathcal{D}_j$ are (or can be made). One popular working assumption is that the distribution of each class has relatively *low-dimensional* intrinsic structures. There are several reasons why this assumption is plausible: 1). High dimensional data are highly redundant; 2). Data that belong to the same class should be similar and correlated to each other; 3). Typically we only care about equivalent structures of $\boldsymbol{x}$ that are invariant to certain classes of deformation and augmentations. Hence we may assume the distribution $\mathcal{D}_j$ of each class has a support on a low-dimensional submanifold, say $\mathcal{M}_j$ with dimension $d_j \ll D$, and the distribution $\mathcal{D}$ of $\boldsymbol{x}$ is supported on the mixture of those submanifolds, $\mathcal{M} = \cup_{j=1}^k \mathcal{M}_j$, in the high-dimensional ambient space $\mathbb{R}^D$, as illustrated in Figure 1 left.

With the manifold assumption in mind, we want to learn a mapping $\boldsymbol{z} = f(\boldsymbol{x}, \theta)$ that maps each of the submanifolds $\mathcal{M}_j \subset \mathbb{R}^D$ to a *linear* subspace $\mathcal{S}_j \subset \mathbb{R}^d$ (see Figure 1 middle). To do so, we require our learned representation to have the following properties:

1. *Between-Class Discriminative:* Features of samples from different classes/clusters should be highly *uncorrelated* and belong to different low-dimensional linear subspaces.

2. *Within-Class Compressible:* Features of samples from the same class/cluster should be relatively *correlated* in a sense that they belong to a low-dimensional linear subspace.

3. *Maximally Diverse Representation:* Dimension (or variance) of features for each class/cluster should be *as large as possible* as long as they stay uncorrelated from the other classes.

Notice that, although the intrinsic structures of each class/cluster may be low-dimensional, they are by no means simply linear in their original representation $\boldsymbol{x}$. Here the subspaces $\{\mathcal{S}_j\}$ can be viewed as nonlinear *generalized principal components* for $\boldsymbol{x}$ [VMS16]. Furthermore, for many clustering or classification tasks (such as object recognition), we consider two samples as *equivalent* if they differ by certain class of domain deformations or augmentations $\mathcal{T} = \{\tau\}$. Hence, we are only interested in low-dimensional structures that are *invariant* to such deformations (i.e., $\boldsymbol{x} \in \mathcal{M}$ iff $\tau(\boldsymbol{x}) \in \mathcal{M}$ for all $\tau \in \mathcal{T}$), which are known to have sophisticated geometric and topological structures [WDCB05] and can be difficult to learn in a principled manner even with CNNs [CW16, CGW19]. There are previous attempts to directly enforce subspace structures on features learned by a deep network for supervised [LQMS18] or unsupervised learning [JZL+17, ZJH+18, PFX+17, ZHF18, ZJH+19, ZLY+19, LQMS18]. However, the *self-expressive* property of subspaces exploited by [JZL+17] does not enforce all the desired properties listed above [HYV20]; [LQMS18] uses a nuclear norm based geometric loss to enforce orthogonality between classes, but does not promote diversity in the learned representations, as we will soon see. Figure 1 right illustrates a representation learned by our method on the CIFAR10 dataset. More details can be found in the experimental Section 3.

## 2 Technical Approach and Method

### 2.1 Measure of Compactness for a Representation

Although the above properties are all highly desirable for the latent representation $\boldsymbol{z}$, they are by no means easy to obtain: Are these properties compatible so that we can expect to achieve them all at

once? If so, is there a *simple but principled* objective that can measure the goodness of the resulting representations in terms of all these properties? The key to these questions is to find a principled "measure of compactness" for the distribution of a random variable $z$ or from its finite samples $Z$. Such a measure should directly and accurately characterize intrinsic geometric or statistical properties of the distribution, in terms of its intrinsic dimension or volume. Unlike cross-entropy (1) or information bottleneck (2), such a measure should not depend explicitly on class labels so that it can work in all supervised, self-supervised, semi-supervised, and unsupervised settings.

**Low-dimensional degenerate distributions.** In information theory [CT06], the notion of entropy $H(z)$ is designed to be such a measure. However, entropy is not well-defined for continuous random variables with degenerate distributions. The same difficulty resides with evaluating mutual information $I(x, z)$ for degenerate distributions. This is unfortunately the case here. To alleviate this difficulty, another related concept in information theory, more specifically in lossy data compression, that measures the "compactness" of a random distribution is the so-called *rate distortion* [CT06]: Given a random variable $z$ and a prescribed precision $\epsilon > 0$, the rate distortion $R(z, \epsilon)$ is the minimal number of binary bits needed to encode $z$ such that the expected decoding error is less than $\epsilon$, i.e., the decoded $\hat{z}$ satisfies $\mathbb{E}[\|z - \hat{z}\|_2] \leq \epsilon$. Although this framework has been successful in explaining feature selection in deep networks [MWHK19], the rate distortion of a random variable is difficult, if not impossible to compute, except for simple distributions such as discrete and Gaussian.

**Nonasymptotic rate distortion for finite samples.** When evaluating the lossy coding rate $R$, one practical difficulty is that we normally do not know the distribution of $z$. Instead, we have a finite number of samples as learned representations where $z_i = f(x_i, \theta) \in \mathbb{R}^d, i = 1, \ldots, m$, for the given data samples $X = [x_1, \ldots, x_m]$. Fortunately, [MDHW07] provides a precise estimate on the number of binary bits needed to encoded finite samples from a subspace-like distribution. In order to encode the learned representation $Z = [z_1, \ldots, z_m]$ up to a precision $\epsilon$, the total number of bits needed is given by the following expression: $\mathcal{L}(Z, \epsilon) \doteq \left(\frac{m+d}{2}\right) \log \det \left(I + \frac{d}{m\epsilon^2} ZZ^\top\right)$. This formula can be derived either by packing $\epsilon$-balls into the space spanned by $Z$ or by computing the number of bits needed to quantize the SVD of $Z$ subject to the precision, see [MDHW07] for proofs. Therefore, the compactness of learned features *as a whole* can be measured in terms of the average coding length per sample (as the sample size $m$ is large), a.k.a. the *coding rate* subject to the distortion $\epsilon$:

$$R(Z, \epsilon) \doteq \frac{1}{2} \log \det \left(I + \frac{d}{m\epsilon^2} ZZ^\top\right). \tag{4}$$

**Rate distortion of data with a mixed distribution.** In general, the features $Z$ of multi-class data may belong to multiple low-dimensional subspaces. To evaluate the rate distortion of such mixed data *more accurately*, we may partition the data $Z$ into multiple subsets: $Z = Z_1 \cup \cdots \cup Z_k$, with each in one low-dim subspace. So the above coding rate (4) is accurate for each subset. For convenience, let $\Pi = \{\Pi_j \in \mathbb{R}^{m \times m}\}_{j=1}^k$ be a set of diagonal matrices whose diagonal entries encode the membership of the $m$ samples in the $k$ classes. More specifically, the diagonal entry $\Pi_j(i, i)$ of $\Pi_j$ indicates the probability of sample $i$ belonging to subset $j$. Therefore $\Pi$ lies in a simplex: $\Omega \doteq \{\Pi \mid \Pi_j \geq 0, \ \Pi_1 + \cdots + \Pi_k = I\}$. Then, according to [MDHW07], with respect to this partition, the average number of bits per sample (the coding rate) is

$$R^c(Z, \epsilon \mid \Pi) \doteq \sum_{j=1}^k \frac{\text{tr}(\Pi_j)}{2m} \log \det \left(I + \frac{d}{\text{tr}(\Pi_j)\epsilon^2} Z\Pi_j Z^\top\right). \tag{5}$$

When $Z$ is given, $R^c(Z, \epsilon \mid \Pi)$ is a concave function of $\Pi$. The function $\log \det(\cdot)$ in the above expressions has been long known as an effective heuristic for rank minimization problems, with guaranteed convergence to local minimum [FHB03]. As it nicely characterizes the rate distortion of Gaussian or subspace-like distributions, $\log \det(\cdot)$ can be very effective in clustering or classification of mixed data [MDHW07, WTL$^+$08, KPCC15].

## 2.2  Principle of Maximal Coding Rate Reduction

On one hand, for learned features to be discriminative, features of different classes/clusters are preferred to be *maximally incoherent* to each other. Hence they together should span a space of the largest possible volume (or dimension) and the coding rate of the whole set $Z$ should be as large as possible. On the other hand, learned features of the same class/cluster should be highly correlated and

coherent. Hence, each class/cluster should only span a space (or subspace) of a very small volume and the coding rate should be as small as possible. Therefore, a good representation $\boldsymbol{Z}$ of $\boldsymbol{X}$ is one such that, given a partition $\boldsymbol{\Pi}$ of $\boldsymbol{Z}$, achieves a large difference between the coding rate for the whole and that for all the subsets:

$$\Delta R(\boldsymbol{Z}, \boldsymbol{\Pi}, \epsilon) \doteq R(\boldsymbol{Z}, \epsilon) - R^c(\boldsymbol{Z}, \epsilon \mid \boldsymbol{\Pi}). \tag{6}$$

If we choose our feature mapping $\boldsymbol{z} = f(\boldsymbol{x}, \theta)$ to be a deep neural network, the overall process of the feature representation and the resulting rate reduction w.r.t. certain partition $\boldsymbol{\Pi}$ can be illustrated by the following diagram:

$$\boldsymbol{X} \xrightarrow{f(\boldsymbol{x}, \theta)} \boldsymbol{Z}(\theta) \xrightarrow{\boldsymbol{\Pi}, \epsilon} \Delta R(\boldsymbol{Z}(\theta), \boldsymbol{\Pi}, \epsilon). \tag{7}$$

Note that $\Delta R$ is *monotonic* in the scale of the features $\boldsymbol{Z}$. So to make the amount of reduction comparable between different representations, we need to *normalize the scale* of the learned features, either by imposing the Frobenius norm of each class $\boldsymbol{Z}_j$ to scale with the number of features in $\boldsymbol{Z}_j \in \mathbb{R}^{d \times m_j}$: $\|\boldsymbol{Z}_j\|_F^2 = m_j$ or by normalizing each feature to be on the unit sphere: $\boldsymbol{z}_i \in \mathbb{S}^{d-1}$. This formulation offers a natural justification for the need of "batch normalization" in the practice of training deep neural networks [IS15]. An alternative, arguably simpler, way to normalize the scale of learned representations is to ensure that the mapping of each layer of the network is approximately *isometric* [QYW+20].

Once the representations are comparable, our goal becomes to learn a set of features $\boldsymbol{Z}(\theta) = f(\boldsymbol{X}, \theta)$ and their partition $\boldsymbol{\Pi}$ (if not given in advance) such that they maximize the reduction between the coding rate of all features and that of the sum of features w.r.t. their classes:

$$\max_{\theta, \boldsymbol{\Pi}} \Delta R(\boldsymbol{Z}(\theta), \boldsymbol{\Pi}, \epsilon) = R(\boldsymbol{Z}(\theta), \epsilon) - R^c(\boldsymbol{Z}(\theta), \epsilon \mid \boldsymbol{\Pi}), \quad \text{s.t.} \quad \|\boldsymbol{Z}_j(\theta)\|_F^2 = m_j, \boldsymbol{\Pi} \in \Omega. \tag{8}$$

We refer to this as the principle of *maximal coding rate reduction* (MCR$^2$), an embodiment of Aristotle's famous quote: "*the whole is greater than the sum of the parts.*" Note that for the clustering purpose alone, one may only care about the sign of $\Delta R$ for deciding whether to partition the data or not, which leads to the greedy algorithm in [MDHW07]. More specifically, in the context of clustering *finite* samples, one needs to use the more precise measure of the coding length mentioned earlier, see [MDHW07] for more details. Here to seek or learn the best representation, we further desire the whole is *maximally* greater than its parts.

**Relationship to information gain.** The maximal coding rate reduction can be viewed as a generalization to *Information Gain* (IG), which aims to maximize the reduction of entropy of a random variable, say $\boldsymbol{z}$, with respect to an observed attribute, say $\boldsymbol{\pi}$: $\max_{\boldsymbol{\pi}} \text{IG}(\boldsymbol{z}, \boldsymbol{\pi}) \doteq H(\boldsymbol{z}) - H(\boldsymbol{z} \mid \boldsymbol{\pi})$, i.e., the *mutual information* between $\boldsymbol{z}$ and $\boldsymbol{\pi}$ [CT06]. Maximal information gain has been widely used in areas such as decision trees [Qui86]. However, MCR$^2$ is used differently in several ways: 1) One typical setting of MCR$^2$ is when the data class labels are given, i.e. $\boldsymbol{\Pi}$ is known, MCR$^2$ focuses on learning representations $\boldsymbol{z}(\theta)$ rather than fitting labels. 2) In traditional settings of IG, the number of attributes in $\boldsymbol{z}$ cannot be so large and their values are discrete (typically binary). Here the "attributes" $\boldsymbol{\Pi}$ represent the probability of a multi-class partition for all samples and their values can even be continuous. 3) As mentioned before, entropy $H(\boldsymbol{z})$ or mutual information $I(\boldsymbol{z}, \boldsymbol{\pi})$ [HFLM+18] is not well-defined for degenerate continuous distributions whereas the rate distortion $R(\boldsymbol{z}, \epsilon)$ is and can be accurately and efficiently computed for (mixed) subspaces, at least.

### 2.3 Properties of the Rate Reduction Function

In theory, the MCR$^2$ principle (8) benefits from great generalizability and can be applied to representations $\boldsymbol{Z}$ of *any* distributions with *any* attributes $\boldsymbol{\Pi}$ as long as the rates $R$ and $R^c$ for the distributions can be accurately and efficiently evaluated. The optimal representation $\boldsymbol{Z}^*$ and partition $\boldsymbol{\Pi}^*$ should have some interesting geometric and statistical properties. We here reveal nice properties of the optimal representation with the special case of subspaces, which have many important use cases in machine learning. When the desired representation for $\boldsymbol{Z}$ is multiple subspaces, the rates $R$ and $R^c$ in (8) are given by (4) and (5), respectively. At the maximal rate reduction, MCR$^2$ achieves its optimal representations, denoted as $\boldsymbol{Z}^* = \boldsymbol{Z}_1^* \cup \cdots \cup \boldsymbol{Z}_k^* \subset \mathbb{R}^d$ with $\mathsf{rank}(\boldsymbol{Z}_j^*) \leq d_j$. One can show that $\boldsymbol{Z}^*$ has the following desired properties (see Appendix A for a formal statement and detailed proofs).

**Theorem 2.1** (Informal Statement). *Suppose $\boldsymbol{Z}^* = \boldsymbol{Z}_1^* \cup \cdots \cup \boldsymbol{Z}_k^*$ is the optimal solution that maximizes the rate reduction* (8). *We have:*

- Between-class Discriminative*: As long as the ambient space is adequately large ($d \geq \sum_{j=1}^k d_j$), the subspaces are all orthogonal to each other,* i.e. $(\boldsymbol{Z}_i^*)^\top \boldsymbol{Z}_j^* = \boldsymbol{0}$ *for $i \neq j$.*

- Maximally Diverse Representation*: As long as the coding precision is adequately high, i.e., $\epsilon^4 < \min_j \left\{ \frac{m_j}{m} \frac{d^2}{d_j^2} \right\}$, each subspace achieves its maximal dimension, i.e.* $\mathsf{rank}(\boldsymbol{Z}_j^*) = d_j$. *In addition, the largest $d_j - 1$ singular values of $\boldsymbol{Z}_j^*$ are equal.*

In other words, in the case of subspaces, the MCR$^2$ principle promotes embedding of data into multiple independent subspaces, with features distributed *isotropically* in each subspace (except for possibly one dimension). In addition, among all such discriminative representations, it prefers the one with the highest dimensions in the ambient space. This is substantially different from the objective of information bottleneck (2).

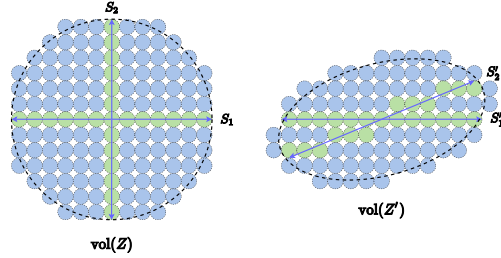

**Comparison to the geometric OLE loss.** To encourage the learned features to be uncorrelated between classes, the work of [LQMS18] has proposed to maximize the difference between the nuclear norm of the whole $\boldsymbol{Z}$ and its subsets $\boldsymbol{Z}_j$, called the *orthogonal low-rank embedding* (OLE) loss: $\max_\theta \mathrm{OLE}(\boldsymbol{Z}(\theta), \boldsymbol{\Pi}) \doteq$

Figure 2: Comparison of two learned representations $\boldsymbol{Z}$ and $\boldsymbol{Z}'$ via reduced rates: $R$ is the number of $\epsilon$-balls packed in the joint distribution and $R^c$ is the sum of the numbers for all the subspaces (the green balls). $\Delta R$ is their difference (the number of blue balls). The MCR$^2$ principle prefers $\boldsymbol{Z}$ (the left one).

$\|\boldsymbol{Z}(\theta)\|_* - \sum_{j=1}^k \|\boldsymbol{Z}_j(\theta)\|_*$, added as a regularizer to the cross-entropy loss (1). The nuclear norm $\|\cdot\|_*$ is a *nonsmooth convex* surrogate for low-rankness and the nonsmoothness potentially poses additional difficulties in using this loss to learn features via gradient descent, whereas $\log \det(\cdot)$ is *smooth concave* instead. Unlike the rate reduction $\Delta R$, OLE is always *negative* and achieves the maximal value 0 when the subspaces are orthogonal, regardless of their dimensions. So in contrast to $\Delta R$, this loss serves as a geometric heuristic and does not promote diverse representations. In fact, OLE typically promotes learning one-dim representations per class, whereas MCR$^2$ encourages learning subspaces with maximal dimensions (Figure 7 of [LQMS18] versus our Figure 6).

**Relation to contrastive learning.** If samples are *evenly* drawn from $k$ classes, a randomly chosen pair $(\boldsymbol{x}_i, \boldsymbol{x}_j)$ is of high probability belonging to difference classes if $k$ is large. For example, when $k \geq 100$, a random pair is of probability 99% belonging to different classes. We may view the learned features of two samples together with their augmentations $\boldsymbol{Z}_i$ and $\boldsymbol{Z}_j$ as two classes. Then the rate reduction $\Delta R_{ij} = R(\boldsymbol{Z}_i \cup \boldsymbol{Z}_j, \epsilon) - \frac{1}{2}(R(\boldsymbol{Z}_i, \epsilon) + R(\boldsymbol{Z}_j, \epsilon))$ gives a "distance" measure for how far the two sample sets are. We may try to further "expand" pairs that likely belong to different classes. From Theorem 2.1, the (averaged) rate reduction $\Delta R_{ij}$ is maximized when features from different samples are uncorrelated $\boldsymbol{Z}_i^\top \boldsymbol{Z}_j = \boldsymbol{0}$ (see Figure 2) and features $\boldsymbol{Z}_i$ from the same sample are highly correlated. Hence, when applied to sample pairs, MCR$^2$ naturally conducts the so-called *contrastive learning* [HCL06, OLV18, HFW$^+$19]. But MCR$^2$ is *not* limited to expand (or compress) pairs of samples and can uniformly conduct "contrastive learning" for a subset with *any number* of samples as long as we know they likely belong to different (or the same) classes, say by randomly sampling subsets from a large number of classes or with a good clustering method.

## 3  Experiments with Instantiations of MCR$^2$

Our theoretical analysis above shows how the *maximal coding rate reduction* (MCR$^2$) is a principled measure for learning discriminative and diverse representations for mixed data. In this section, we demonstrate experimentally how this principle alone, *without any other heuristics,* is adequate to learning good representations in the supervised, self-supervised, and unsupervised learning settings in a unified fashion. Our goal here is to validate effectiveness of this principle through its most basic usage and fair comparison with existing frameworks. More implementation details and experiments are given in Appendix B. The code can be found in `https://github.com/ryanchankh/mcr2`.

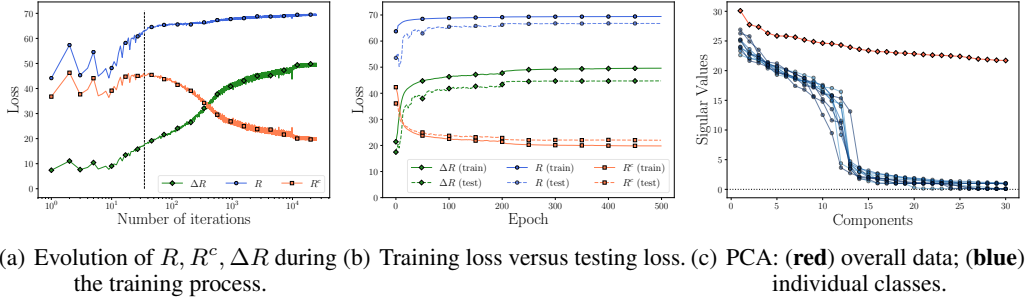

(a) Evolution of $R, R^c, \Delta R$ during the training process.  (b) Training loss versus testing loss.  (c) PCA: (**red**) overall data; (**blue**) individual classes.

Figure 3: Evolution of the rates of MCR$^2$ in the training process and principal components of learned features.

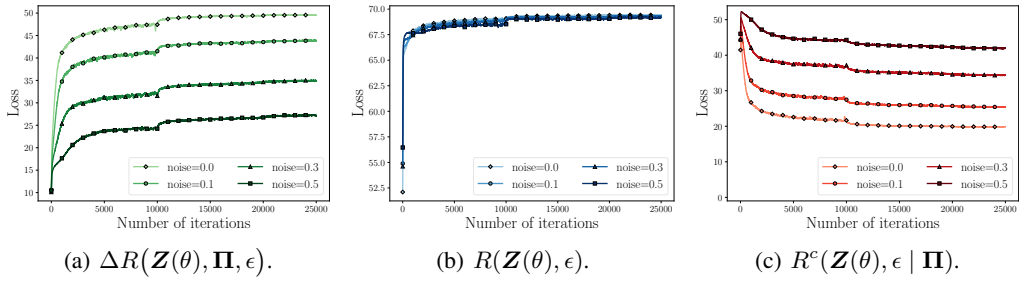

(a) $\Delta R\big(\boldsymbol{Z}(\theta), \boldsymbol{\Pi}, \epsilon\big)$.  (b) $R(\boldsymbol{Z}(\theta), \epsilon)$.  (c) $R^c(\boldsymbol{Z}(\theta), \epsilon \mid \boldsymbol{\Pi})$.

Figure 4: Evolution of rates $R, R^c, \Delta R$ of MCR$^2$ during training with corrupted labels.

## 3.1 Supervised Learning of Robust Discriminative Features

**Supervised learning via rate reduction.** When class labels are provided during training, we assign the membership (diagonal) matrix $\boldsymbol{\Pi} = \{\boldsymbol{\Pi}_j\}_{j=1}^k$ as follows: for each sample $\boldsymbol{x}_i$ with label $j$, set $\boldsymbol{\Pi}_j(i,i) = 1$ and $\boldsymbol{\Pi}_l(i,i) = 0, \forall l \neq j$. Then the mapping $f(\cdot, \theta)$ can be learned by optimizing (8), where $\boldsymbol{\Pi}$ remains constant. We apply stochastic gradient descent to optimize MCR$^2$, and for each iteration we use mini-batch data $\{(\boldsymbol{x}_i, \boldsymbol{y}_i)\}_{i=1}^m$ to approximate the MCR$^2$ loss.

**Evaluation via classification.** As we will see, in the supervised setting, the learned representation has very clear subspace structures. So to evaluate the learned representations, we consider a natural nearest subspace classifier. For each class of learned features $\boldsymbol{Z}_j$, let $\boldsymbol{\mu}_j \in \mathbb{R}^p$ be its mean and $\boldsymbol{U}_j \in \mathbb{R}^{p \times r_j}$ be the first $r_j$ principal components for $\boldsymbol{Z}_j$, where $r_j$ is the estimated dimension of class $j$. The predicted label of a test data $\boldsymbol{x}'$ is given by $j' = \arg\min_{j \in \{1,\dots,k\}} \|(\boldsymbol{I} - \boldsymbol{U}_j \boldsymbol{U}_j^\top)(f(\boldsymbol{x}', \theta) - \boldsymbol{\mu}_j)\|_2^2$.

**Experiments on real data.** We consider CIFAR10 dataset [Kri09] and ResNet-18 [HZRS16] for $f(\cdot, \theta)$. We replace the last linear layer of ResNet-18 by a two-layer fully connected network with ReLU activation function such that the output dimension is 128. We set the mini-batch size as $m = 1,000$ and the precision parameter $\epsilon^2 = 0.5$. More results can be found in Appendix B.3.2.

Figure 3(a) illustrates how the two rates and their difference (for both training and test data) evolves over epochs of training: After an initial phase, $R$ gradually increases while $R^c$ decreases, indicating that features $\boldsymbol{Z}$ are expanding as a whole while each class $\boldsymbol{Z}_j$ is being compressed. Figure 3(c) shows the distribution of singular values per $\boldsymbol{Z}_j$ and Figure 1 (right) shows the angles of features sorted by class. Compared to the geometric loss [LQMS18], our features are *not only orthogonal but also of much higher dimension*. We compare the singular values of representations, both overall data and individual classes, learned by using cross-entropy and MCR$^2$ in Figure 6 and Figure 7 in Appendix B.3.1. We find that the representations learned by using MCR$^2$ loss are much more diverse than the ones learned by using cross-entropy loss. In addition, we find that we are able to select diverse images from the same class according to the "principal" components of the learned features (see Figure 8 and Figure 9 in Appendix B.3.1).

**Robustness to corrupted labels.** Because MCR$^2$ by design encourages richer representations that preserves intrinsic structures from the data $\boldsymbol{X}$, training relies less on class labels than traditional loss such as cross-entropy (CE). To verify this, we train the same network using both CE and MCR$^2$ with

certain ratios of *randomly corrupted* training labels. Figure 4 illustrates the learning process: for different levels of corruption, while the rate for the whole set always converges to the same value, the rates for the classes are inversely proportional to the ratio of corruption, indicating our method only compresses samples with valid labels. The classification results are summarized in Table 1. By applying *exact the same* training parameters, MCR$^2$ is significantly more robust than CE, especially with higher ratio of corrupted labels. This can be an advantage in the settings of self-supervised learning or constrastive learning when the grouping information can be very noisy. More detailed comparison between MCR$^2$ and OLE [LQMS18], Large Margin Deep Networks [EKM$^+$18], and ITLM [SS19] on learning from noisy labels can be found in Appendix B.4 (Table 7).

Table 1: Classification results with features learned with labels corrupted at different levels.

|  | RATIO=0.1 | RATIO=0.2 | RATIO=0.3 | RATIO=0.4 | RATIO=0.5 |
|---|---|---|---|---|---|
| CE TRAINING | 90.91% | 86.12% | 79.15% | 72.45% | 60.37% |
| MCR$^2$ TRAINING | **91.16%** | **89.70%** | **88.18%** | **86.66%** | **84.30%** |

## 3.2 Self-supervised Learning of Invariant Features

**Learning invariant features via rate reduction.** Motivated by self-supervised learning algorithms [LHB04, KRFL09, OLV18, HFW$^+$19, WXYL18], we use the MCR$^2$ principle to learn representations that are *invariant* to certain class of transformations/augmentations, say $\mathcal{T}$ with a distribution $P_{\mathcal{T}}$. Given a mini-batch of data $\{\boldsymbol{x}_j\}_{j=1}^k$, we augment each sample $\boldsymbol{x}_j$ with $n$ transformations/augmentations $\{\tau_i(\cdot)\}_{i=1}^n$ randomly drawn from $P_{\mathcal{T}}$. We simply label all the augmented samples $\boldsymbol{X}_j = [\tau_1(\boldsymbol{x}_j), \ldots, \tau_n(\boldsymbol{x}_j)]$ of $\boldsymbol{x}_j$ as the $j$-th class, and $\boldsymbol{Z}_j$ the corresponding learned features. Using this self-labeled data, we train our feature mapping $f(\cdot, \theta)$ the same way as the supervised setting above. For every mini-batch, the total number of samples for training is $m = kn$.

**Evaluation via clustering.** To learn invariant features, our formulation itself does *not* require the original samples $\boldsymbol{x}_j$ come from a fixed number of classes. For evaluation, we may train on a few classes and observe how the learned features facilitate classification or clustering of the data. A common method to evaluate learned features is to train an additional linear classifier [OLV18, HFW$^+$19], with ground truth labels. But for our purpose, because we explicitly verify whether the so-learned invariant features have good subspace structures when the samples come from $k$ classes, we use an off-the-shelf subspace clustering algorithm EnSC [YLRV16], which is computationally efficient and is provably correct for data with well-structured subspaces. We also use K-Means on the original data $\boldsymbol{X}$ as our baseline for comparison. We use normalized mutual information (NMI), clustering accuracy (ACC), and adjusted rand index (ARI) for our evaluation metrics, see Appendix B.4.2 for their detailed definitions.

**Controlling dynamics of expansion and compression.** By directly optimizing the rate reduction $\Delta R = R - R^c$, we achieve 0.570 clustering accuracy on CIFAR10 dataset, which is the second best result compared with previous methods. More details can be found in Appendix B.4.1. Empirically, we observe that, without class labels, the overall *coding rate* $R$ expands quickly and the MCR$^2$ loss saturates (at a local maximum), see Fig 5(a). Our experience suggests that learning a good representation from unlabeled data might be too ambitious when directly optimizing the original $\Delta R$. Nonetheless, from the geometric meaning of $R$ and $R^c$, one can design a different learning strategy by controlling the dynamics of expansion and compression differently during training. For instance, we may re-scale the rate by replacing $R(\boldsymbol{Z}, \epsilon)$ with $\widetilde{R}(\boldsymbol{Z}, \epsilon) \doteq \frac{1}{2\gamma_1} \log \det(\boldsymbol{I} + \frac{\gamma_2 d}{m\epsilon^2} \boldsymbol{Z}\boldsymbol{Z}^\top)$. With $\gamma_1 = \gamma_2 = k$, the learning dynamics change from Fig 5(a) to Fig 5(b): All features are first compressed then gradually expand. We denote the controlled MCR$^2$ training by MCR$^2$-CTRL.

**Experiments on real data.** Similar to the supervised learning setting, we train *exactly the same* ResNet-18 network on the CIFAR10, CIFAR100, and STL10 [CNL11] datasets. We set the mini-batch size as $k = 20$, number of augmentations for each sample as $n = 50$ and the precision parameter as $\epsilon^2 = 0.5$. Table 2 shows the results of the proposed MCR$^2$-CTRL in comparison with methods JULE [YPB16], RTM [NMM19], DEC [XGF16], DAC [CWM$^+$17], and DCCM [WLW$^+$19] that have achieved the best results on these datasets. Surprisingly, without utilizing any inter-class or inter-sample information and heuristics on the data, the invariant features learned by our method with augmentations alone achieves a better performance over other highly engineered clustering methods. More comparisons and ablation studies can be found in Appendix B.4.2.

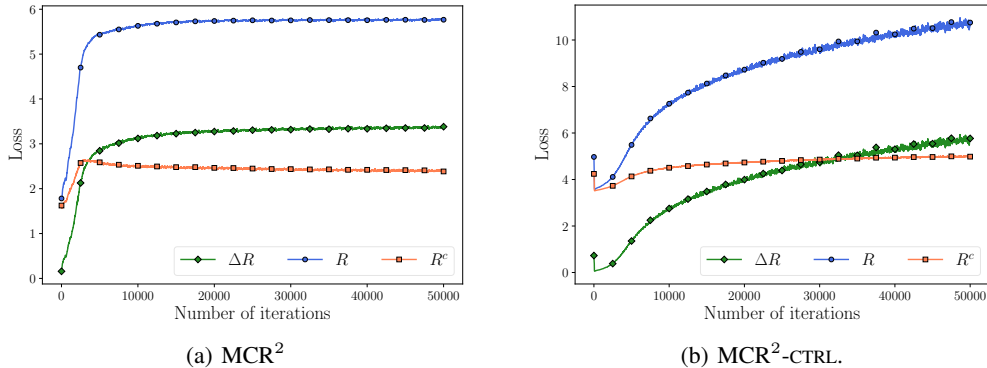

Figure 5: Evolution of the rates of (**left**) MCR$^2$ and (**right**) MCR$^2$-CTRL in the training process in the self-supervised setting on CIFAR10 dataset.

Table 2: Clustering results on CIFAR10, CIFAR100, and STL10 datasets.

| DATASET | METRIC | K-MEANS | JULE | RTM | DEC | DAC | DCCM | MCR$^2$-CTRL |
|---------|--------|---------|------|-----|-----|-----|------|--------------|
| CIFAR10 | NMI | 0.087 | 0.192 | 0.197 | 0.257 | 0.395 | 0.496 | **0.630** |
| | ACC | 0.229 | 0.272 | 0.309 | 0.301 | 0.521 | 0.623 | **0.684** |
| | ARI | 0.049 | 0.138 | 0.115 | 0.161 | 0.305 | 0.408 | **0.508** |
| CIFAR100 | NMI | 0.084 | 0.103 | - | 0.136 | 0.185 | 0.285 | **0.387** |
| | ACC | 0.130 | 0.137 | - | 0.185 | 0.237 | 0.327 | **0.375** |
| | ARI | 0.028 | 0.033 | - | 0.050 | 0.087 | 0.173 | **0.178** |
| STL10 | NMI | 0.124 | 0.182 | - | 0.276 | 0.365 | 0.376 | **0.446** |
| | ACC | 0.192 | 0.182 | - | 0.359 | 0.470 | 0.482 | **0.491** |
| | ARI | 0.061 | 0.164 | - | 0.186 | 0.256 | 0.262 | **0.290** |

Nevertheless, compared to the representations learned in the supervised setting where the optimal partition $\Pi$ in (8) is initialized by correct class information, the representations here learned with self-supervised classes are far from being optimal. It remains wide open how to design better optimization strategies and dynamics to learn from unlabelled or partially-labelled data better representations (and the associated partitions) close to the global maxima of the MCR$^2$ objective (8).

## 4   Conclusion and Future Work

This work provides rigorous theoretical justifications and clear empirical evidences for why the maximal coding rate reduction (MCR$^2$) is a fundamental principle for learning discriminative low-dim representations in almost all learning settings. It unifies and explains existing effective frameworks and heuristics widely practiced in the (deep) learning literature. It remains open *why* MCR$^2$ is robust to label noises in the supervised setting, *why* self-learned features with MCR$^2$ alone are effective for clustering, and *how* in future practice instantiations of this principle can be systematically harnessed to further improve clustering or classification tasks.

We believe that MCR$^2$ gives a principled and practical objective for (deep) learning and can potentially lead to better design operators and architectures of a deep network. A potential direction is to monitor quantitatively the amount of rate reduction $\Delta R$ gained through every layer of the deep network. By optimizing the rate reduction through the network layers, it is no longer engineered as a "black box."

On the learning theoretical aspect, although this work has demonstrated only with mixed subspaces, this principle applies to any mixed distributions or structures, for which configurations that achieve maximal rate reduction are of independent theoretical interest. Another interesting note is that the MCR$^2$ formulation goes beyond the supervised multi-class learning setting often studied through empirical risk minimization (ERM) [DSBDSS15]. It is more related to the expectation maximization (EMX) framework [BDHM$^+$17], in which the notion of "compression" plays a crucial role for purely theoretical analysis. We hope this work provides a good connection between machine learning theory and its practice.

## Broader Impact

The principle proposed in this work aims to maximally capture the intrinsic structures of the data that justify meaningful classification of clustering of real-world data. Our framework discourages models from learning by only fitting or overfitting the labeled data with a black box, enables us to identify the intrinsic structures of the data hence the true causes for meaningful classification or clustering.

This methodology also allows us to maximally reduce the effects of bias or even mistakes that might be introduced in the labeled data. We believe this is the basis for truly interpretable (deep) learning, and hence the basis for developing truly robust and fair machine learning algorithms and systems, with clear performance guarantees.

## Acknowledgments and Disclosure of Funding

Yi acknowledges support from ONR grant N00014-20-1-2002 and the joint Simons Foundation-NSF DMS grant #2031899, as well as support from Berkeley FHL Vive Center for Enhanced Reality and Berkeley Center for Augmented Cognition. Chong, Chaobing, and Yi acknowledge support from Tsinghua-Berkeley Shenzhen Institute (TBSI) Research Fund. Yaodong and Yi like to acknowledge support from Berkeley AI Research (BAIR).

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
