[Supplementary Material]

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

# Appendices

## A  Properties of the Rate Reduction Function

This section is organized as follows. We present background and preliminary results for the $\log\det(\cdot)$ function and the coding rate function in Section A.1. Then, Section A.2 and A.3 provide technical lemmas for bounding the coding rate and coding rate reduction functions, respectively. Such lemmas are key results for proving our main theoretical results, which are stated informally in Theorem 2.1 and formally in Section A.4. Finally, proof of our main theoretical results is provided in Section A.5.

**Notations**  Throughout this section, we use $\mathbb{S}^d_{++}$, $\mathbb{R}_+$ and $\mathbb{Z}_{++}$ to denote the set of symmetric positive definite matrices of size $d \times d$, nonnegative real numbers and positive integers, respectively.

### A.1  Preliminaries

**Properties of the $\log\det(\cdot)$ function.**

**Lemma A.1.** *The function* $\log\det(\cdot) : \mathbb{S}^d_{++} \to \mathbb{R}$ *is strictly concave. That is,*

$$\log\det((1-\alpha)\boldsymbol{Z}_1 + \alpha\boldsymbol{Z}_2)) \geq (1-\alpha)\log\det(\boldsymbol{Z}_1) + \alpha\log\det(\boldsymbol{Z}_2)$$

*for any* $\alpha \in (0,1)$ *and* $\{\boldsymbol{Z}_1, \boldsymbol{Z}_2\} \subseteq \mathbb{S}^d_{++}$, *with equality holds if and only if* $\boldsymbol{Z}_1 = \boldsymbol{Z}_2$.

*Proof.*  Consider an arbitrary line given by $\boldsymbol{Z} = \boldsymbol{Z}_0 + t\Delta\boldsymbol{Z}$ where $\boldsymbol{Z}_0$ and $\Delta\boldsymbol{Z} \neq \boldsymbol{0}$ are symmetric matrices of size $d \times d$. Let $f(t) \doteq \log\det(\boldsymbol{Z}_0 + t\Delta\boldsymbol{Z})$ be a function defined on an interval of values of $t$ for which $\boldsymbol{Z}_0 + t\Delta\boldsymbol{Z} \in \mathbb{S}^d_{++}$. Following the same argument as in [BV04], we may assume $\boldsymbol{Z}_0 \in \mathbb{S}^d_{++}$ and get

$$f(t) = \log\det\boldsymbol{Z}_0 + \sum_{i=1}^{d}\log(1 + t\lambda_i),$$

where $\{\lambda_i\}_{i=1}^{d}$ are eigenvalues of $\boldsymbol{Z}_0^{-\frac{1}{2}}\Delta\boldsymbol{Z}\boldsymbol{Z}_0^{-\frac{1}{2}}$. The second order derivative of $f(t)$ is given by

$$f''(t) = -\sum_{i=1}^{d}\frac{\lambda_i^2}{(1 + t\lambda_i)^2} < 0.$$

Therefore, $f(t)$ is strictly concave along the line $\boldsymbol{Z} = \boldsymbol{Z}_0 + t\Delta\boldsymbol{Z}$. By definition, we conclude that $\log\det(\cdot)$ is strictly concave. $\qquad\square$

**Properties of the coding rate function.**  The following properties, also known as the Sylvester's determinant theorem, for the coding rate function are known in the paper [MDHW07].

**Lemma A.2** (Commutative property [MDHW07]). *For any* $\boldsymbol{Z} \in \mathbb{R}^{d \times m}$ *we have*

$$R(\boldsymbol{Z}, \epsilon) \doteq \frac{1}{2}\log\det\left(\boldsymbol{I} + \frac{d}{m\epsilon^2}\boldsymbol{Z}\boldsymbol{Z}^\top\right) = \frac{1}{2}\log\det\left(\boldsymbol{I} + \frac{d}{m\epsilon^2}\boldsymbol{Z}^\top\boldsymbol{Z}\right).$$

**Lemma A.3** (Invariant property [MDHW07]). *For any* $\boldsymbol{Z} \in \mathbb{R}^{d \times m}$ *and any orthogonal matrices* $\boldsymbol{U} \in \mathbb{R}^{d \times d}$ *and* $\boldsymbol{V} \in \mathbb{R}^{m \times m}$ *we have*

$$R(\boldsymbol{Z}, \epsilon) = R(\boldsymbol{U}\boldsymbol{Z}\boldsymbol{V}^\top, \epsilon).$$

### A.2  Lower and Upper Bounds for Coding Rate

The following result provides an upper and a lower bound on the coding rate of $\boldsymbol{Z}$ as a function of the coding rate for its components $\{\boldsymbol{Z}_j\}_{j=1}^{k}$. The lower bound is tight when all the components $\{\boldsymbol{Z}_j\}_{j=1}^{k}$ have the same covariance (assuming that they have zero mean). The upper bound is tight when the components $\{\boldsymbol{Z}_j\}_{j=1}^{k}$ are pair-wise orthogonal.

**Lemma A.4.** *For any* $\{ \boldsymbol{Z}_j \in \mathbb{R}^{d \times m_j} \}_{j=1}^k$ *and any* $\epsilon > 0$, *let* $\boldsymbol{Z} = [\boldsymbol{Z}_1, \cdots, \boldsymbol{Z}_k] \in \mathbb{R}^{d \times m}$ *with* $m = \sum_{j=1}^k m_j$. *We have*

$$\sum_{j=1}^k \frac{m_j}{2} \log \det \left( \boldsymbol{I} + \frac{d}{m_j \epsilon^2} \boldsymbol{Z}_j \boldsymbol{Z}_j^\top \right) \le \frac{m}{2} \log \det \left( \boldsymbol{I} + \frac{d}{m \epsilon^2} \boldsymbol{Z} \boldsymbol{Z}^\top \right)$$

$$\le \sum_{j=1}^k \frac{m}{2} \log \det \left( \boldsymbol{I} + \frac{d}{m \epsilon^2} \boldsymbol{Z}_j \boldsymbol{Z}_j^\top \right), \tag{9}$$

*where the first equality holds if and only if*

$$\frac{\boldsymbol{Z}_1 \boldsymbol{Z}_1^\top}{m_1} = \frac{\boldsymbol{Z}_2 \boldsymbol{Z}_2^\top}{m_2} = \cdots = \frac{\boldsymbol{Z}_k \boldsymbol{Z}_k^\top}{m_k},$$

*and the second equality holds if and only if* $\boldsymbol{Z}_{j_1}^\top \boldsymbol{Z}_{j_2} = \boldsymbol{0}$ *for all* $1 \le j_1 < j_2 \le k$.

*Proof.* By Lemma A.1, $\log \det(\cdot)$ is strictly concave. Therefore,

$$\log \det \left( \sum_{j=1}^k \alpha_j \boldsymbol{S}_j \right) \ge \sum_{j=1}^k \alpha_j \log \det(\boldsymbol{S}_j), \text{ for all } \{\alpha_j > 0\}_{j=1}^k, \sum_{j=1}^k \alpha_j = 1 \text{ and } \{\boldsymbol{S}_j \in \mathbb{S}_{++}^d\}_{j=1}^k,$$

where equality holds if and only if $\boldsymbol{S}_1 = \boldsymbol{S}_2 = \cdots = \boldsymbol{S}_k$. Take $\alpha_j = \frac{m_j}{m}$ and $\boldsymbol{S}_j = \boldsymbol{I} + \frac{d}{m_j \epsilon^2} \boldsymbol{Z}_j \boldsymbol{Z}_j^\top$, we get

$$\frac{m}{2} \log \det \left( \boldsymbol{I} + \frac{d}{m \epsilon^2} \boldsymbol{Z} \boldsymbol{Z}^\top \right) \ge \sum_{j=1}^k \frac{m_j}{2} \log \det \left( \boldsymbol{I} + \frac{d}{m_j \epsilon^2} \boldsymbol{Z}_j \boldsymbol{Z}_j^\top \right),$$

with equality holds if and only if $\frac{\boldsymbol{Z}_1 \boldsymbol{Z}_1^\top}{m_1} = \cdots = \frac{\boldsymbol{Z}_k \boldsymbol{Z}_k^\top}{m_k}$. This proves the lower bound in (9).

We now prove the upper bound. By the strict concavity of $\log \det(\cdot)$, we have

$$\log \det(\boldsymbol{Q}) \le \log \det(\boldsymbol{S}) + \langle \nabla \log \det(\boldsymbol{S}), \boldsymbol{Q} - \boldsymbol{S} \rangle, \text{ for all } \{\boldsymbol{Q}, \boldsymbol{S}\} \subseteq \mathbb{S}_{++}^m,$$

where equality holds if and only if $\boldsymbol{Q} = \boldsymbol{S}$. Plugging in $\nabla \log \det(\boldsymbol{S}) = \boldsymbol{S}^{-1}$ (see e.g., [BV04]) and $\boldsymbol{S}^{-1} = (\boldsymbol{S}^{-1})^\top$ gives

$$\log \det(\boldsymbol{Q}) \le \log \det(\boldsymbol{S}) + \text{tr}(\boldsymbol{S}^{-1} \boldsymbol{Q}) - m. \tag{10}$$

We now take

$$\boldsymbol{Q} = \boldsymbol{I} + \frac{d}{m\epsilon^2} \boldsymbol{Z}^\top \boldsymbol{Z} = \boldsymbol{I} + \frac{d}{m\epsilon^2} \begin{bmatrix} \boldsymbol{Z}_1^\top \boldsymbol{Z}_1 & \boldsymbol{Z}_1^\top \boldsymbol{Z}_2 & \cdots & \boldsymbol{Z}_1^\top \boldsymbol{Z}_k \\ \boldsymbol{Z}_2^\top \boldsymbol{Z}_1 & \boldsymbol{Z}_2^\top \boldsymbol{Z}_2 & \cdots & \boldsymbol{Z}_2^\top \boldsymbol{Z}_k \\ \vdots & \vdots & \ddots & \vdots \\ \boldsymbol{Z}_k^\top \boldsymbol{Z}_1 & \boldsymbol{Z}_k^\top \boldsymbol{Z}_2 & \cdots & \boldsymbol{Z}_k^\top \boldsymbol{Z}_k \end{bmatrix}, \text{ and} \tag{11}$$

$$\boldsymbol{S} = \boldsymbol{I} + \frac{d}{m\epsilon^2} \begin{bmatrix} \boldsymbol{Z}_1^\top \boldsymbol{Z}_1 & \boldsymbol{0} & \cdots & \boldsymbol{0} \\ \boldsymbol{0} & \boldsymbol{Z}_2^\top \boldsymbol{Z}_2 & \cdots & \boldsymbol{0} \\ \vdots & \vdots & \ddots & \vdots \\ \boldsymbol{0} & \boldsymbol{0} & \cdots & \boldsymbol{Z}_k^\top \boldsymbol{Z}_k \end{bmatrix}.$$

From the property of determinant for block diagonal matrix, we have

$$\log \det(\boldsymbol{S}) = \sum_{j=1}^k \log \det \left( \boldsymbol{I} + \frac{d}{m\epsilon^2} \boldsymbol{Z}_j^\top \boldsymbol{Z}_j \right). \tag{12}$$

Also, note that

$$\mathrm{tr}(\boldsymbol{S}^{-1}\boldsymbol{Q})$$

$$= \mathrm{tr} \begin{bmatrix} (\boldsymbol{I} + \frac{d}{m\epsilon^2}\boldsymbol{Z}_1^\top \boldsymbol{Z}_1)^{-1}(\boldsymbol{I} + \frac{d}{m\epsilon^2}\boldsymbol{Z}_1^\top \boldsymbol{Z}_1) & \cdots & (\boldsymbol{I} + \frac{d}{m\epsilon^2}\boldsymbol{Z}_1^\top \boldsymbol{Z}_1)^{-1}(\boldsymbol{I} + \frac{d}{m\epsilon^2}\boldsymbol{Z}_1^\top \boldsymbol{Z}_k) \\ \vdots & \ddots & \vdots \\ (\boldsymbol{I} + \frac{d}{m\epsilon^2}\boldsymbol{Z}_k^\top \boldsymbol{Z}_k)^{-1}(\boldsymbol{I} + \frac{d}{m\epsilon^2}\boldsymbol{Z}_k^\top \boldsymbol{Z}_1) & \cdots & (\boldsymbol{I} + \frac{d}{m\epsilon^2}\boldsymbol{Z}_k^\top \boldsymbol{Z}_k)^{-1}(\boldsymbol{I} + \frac{d}{m\epsilon^2}\boldsymbol{Z}_k^\top \boldsymbol{Z}_k) \end{bmatrix}$$

$$= \mathrm{tr} \begin{bmatrix} \boldsymbol{I} & \cdots & * \\ \vdots & \ddots & \vdots \\ * & \cdots & \boldsymbol{I} \end{bmatrix} = m, \tag{13}$$

where "*" denotes nonzero quantities that are irrelevant for the purpose of computing the trace. Plugging (12) and (13) back in (10) gives

$$\frac{m}{2} \log \det \left( \boldsymbol{I} + \frac{d}{m\epsilon^2} \boldsymbol{Z}^\top \boldsymbol{Z} \right) \le \sum_{j=1}^{k} \frac{m}{2} \log \det \left( \boldsymbol{I} + \frac{d}{m\epsilon^2} \boldsymbol{Z}_j^\top \boldsymbol{Z}_j \right),$$

where the equality holds if and only if $\boldsymbol{Q} = \boldsymbol{S}$, which by the formulation in (11), holds if and only if $\boldsymbol{Z}_{j_1}^\top \boldsymbol{Z}_{j_2} = \boldsymbol{0}$ for all $1 \le j_1 < j_2 \le k$. Further using the result in Lemma A.2 gives

$$\frac{m}{2} \log \det \left( \boldsymbol{I} + \frac{d}{m\epsilon^2} \boldsymbol{Z}\boldsymbol{Z}^\top \right) \le \sum_{j=1}^{k} \frac{m}{2} \log \det \left( \boldsymbol{I} + \frac{d}{m\epsilon^2} \boldsymbol{Z}_j\boldsymbol{Z}_j^\top \right),$$

which produces the upper bound in (9). □

## A.3 An Upper Bound on Coding Rate Reduction

We may now provide an upper bound on the coding rate reduction $\Delta R(\boldsymbol{Z}, \boldsymbol{\Pi}, \epsilon)$ (defined in (8)) in terms of its individual components $\{\boldsymbol{Z}_j\}_{j=1}^{k}$.

**Lemma A.5.** *For any $\boldsymbol{Z} \in \mathbb{R}^{d \times m}, \boldsymbol{\Pi} \in \Omega$ and $\epsilon > 0$, let $\boldsymbol{Z}_j \in \mathbb{R}^{d \times m_j}$ be $\boldsymbol{Z}\boldsymbol{\Pi}_j$ with zero columns removed. We have*

$$\Delta R(\boldsymbol{Z}, \boldsymbol{\Pi}, \epsilon) \le \sum_{j=1}^{k} \frac{1}{2m} \log \left( \frac{\det^m \left( \boldsymbol{I} + \frac{d}{m\epsilon^2} \boldsymbol{Z}_j \boldsymbol{Z}_j^\top \right)}{\det^{m_j} \left( \boldsymbol{I} + \frac{d}{m_j \epsilon^2} \boldsymbol{Z}_j \boldsymbol{Z}_j^\top \right)} \right), \tag{14}$$

*with equality holds if and only if $\boldsymbol{Z}_{j_1}^\top \boldsymbol{Z}_{j_2} = \boldsymbol{0}$ for all $1 \le j_1 < j_2 \le k$.*

*Proof.* From (4), (5) and (6), we have

$$\Delta R(\boldsymbol{Z}, \boldsymbol{\Pi}, \epsilon)$$
$$= R(\boldsymbol{Z}, \epsilon) - R^c(\boldsymbol{Z}, \epsilon \mid \boldsymbol{\Pi})$$
$$= \frac{1}{2} \log \left( \det \left( \boldsymbol{I} + \frac{d}{m\epsilon^2} \boldsymbol{Z}\boldsymbol{Z}^\top \right) \right) - \sum_{j=1}^{k} \left\{ \frac{\mathrm{tr}(\boldsymbol{\Pi}_j)}{2m} \log \left( \det \left( \boldsymbol{I} + d\frac{\boldsymbol{Z}\boldsymbol{\Pi}_j\boldsymbol{Z}^\top}{\mathrm{tr}(\boldsymbol{\Pi}_j)\epsilon^2} \right) \right) \right\}$$
$$= \frac{1}{2} \log \left( \det \left( \boldsymbol{I} + \frac{d}{m\epsilon^2} \boldsymbol{Z}\boldsymbol{Z}^\top \right) \right) - \sum_{j=1}^{k} \left\{ \frac{m_j}{2m} \log \left( \det \left( \boldsymbol{I} + d\frac{\boldsymbol{Z}_j\boldsymbol{Z}_j^\top}{m_j\epsilon^2} \right) \right) \right\}$$
$$\le \sum_{j=1}^{k} \frac{1}{2} \log \left( \det \left( \boldsymbol{I} + \frac{d}{m\epsilon^2} \boldsymbol{Z}_j\boldsymbol{Z}_j^\top \right) \right) - \sum_{j=1}^{k} \left\{ \frac{m_j}{2m} \log \left( \det \left( \boldsymbol{I} + d\frac{\boldsymbol{Z}_j\boldsymbol{Z}_j^\top}{m_j\epsilon^2} \right) \right) \right\}$$
$$= \sum_{j=1}^{k} \frac{1}{2m} \log \left( \det^m \left( \boldsymbol{I} + \frac{d}{m\epsilon^2} \boldsymbol{Z}_j\boldsymbol{Z}_j^\top \right) \right) - \sum_{j=1}^{k} \left\{ \frac{1}{2m} \log \left( \det^{m_j} \left( \boldsymbol{I} + d\frac{\boldsymbol{Z}_j\boldsymbol{Z}_j^\top}{m_j\epsilon^2} \right) \right) \right\}$$
$$= \sum_{j=1}^{k} \frac{1}{2m} \log \left( \frac{\det^m \left( \boldsymbol{I} + \frac{d}{m\epsilon^2} \boldsymbol{Z}_j\boldsymbol{Z}_j^\top \right)}{\det^{m_j} \left( \boldsymbol{I} + \frac{d}{m_j\epsilon^2} \boldsymbol{Z}_j\boldsymbol{Z}_j^\top \right)} \right),$$

where the inequality follows from the upper bound in Lemma A.4, and that the equality holds if and only if $\boldsymbol{Z}_{j_1}^\top \boldsymbol{Z}_{j_2} = \boldsymbol{0}$ for all $1 \le j_1 < j_2 \le k$. □

## A.4 Main Results: Properties of Maximal Coding Rate Reduction

We now present our main theoretical results. The following theorem states that for any fixed encoding of the partition $\mathbf{\Pi}$, the coding rate reduction is maximized by data $\mathbf{Z}$ that is maximally discriminative between different classes and is diverse within each of the classes. This result holds provided that the sum of rank for different classes is small relative to the ambient dimension, and that $\epsilon$ is small.

**Theorem A.6.** *Let* $\mathbf{\Pi} = \{\mathbf{\Pi}_j \in \mathbb{R}^{m \times m}\}_{j=1}^k$ *with* $\{\mathbf{\Pi}_j \geq \mathbf{0}\}_{j=1}^k$ *and* $\mathbf{\Pi}_1 + \cdots + \mathbf{\Pi}_k = \mathbf{I}$ *be a given set of diagonal matrices whose diagonal entries encode the membership of the $m$ samples in the $k$ classes. Given any $\epsilon > 0$, $d > 0$ and $\{d \geq d_j > 0\}_{j=1}^k$, consider the optimization problem*

$$\mathbf{Z}^* \in \underset{\mathbf{Z} \in \mathbb{R}^{d \times m}}{\arg\max} \, \Delta R(\mathbf{Z}, \mathbf{\Pi}, \epsilon)$$

$$\text{s.t. } \|\mathbf{Z}\mathbf{\Pi}_j\|_F^2 = \text{tr}(\mathbf{\Pi}_j), \ \text{rank}(\mathbf{Z}\mathbf{\Pi}_j) \leq d_j, \ \forall j \in \{1, \ldots, k\}. \tag{15}$$

*Under the conditions*

- *(Large ambient dimension)* $d \geq \sum_{j=1}^k d_j$, *and*

- *(High coding precision)* $\epsilon^4 < \min_{j \in \{1,\ldots,k\}} \left\{ \frac{\text{tr}(\mathbf{\Pi}_j)}{m} \frac{d^2}{d_j^2} \right\}$,

*the optimal solution $\mathbf{Z}^*$ satisfies*

- *(Between-class discriminative)* $(\mathbf{Z}_{j_1}^*)^\top \mathbf{Z}_{j_2}^* = \mathbf{0}$ *for all* $1 \leq j_1 < j_2 \leq k$, *i.e.,* $\mathbf{Z}_{j_1}^*$ *and* $\mathbf{Z}_{j_2}^*$ *lie in orthogonal subspaces, and*

- *(Within-class diverse)* *For each* $j \in \{1, \ldots, k\}$, *the rank of* $\mathbf{Z}_j^*$ *is equal to* $d_j$ *and either all singular values of* $\mathbf{Z}_j^*$ *are equal to* $\frac{\text{tr}(\mathbf{\Pi}_j)}{d_j}$, *or the* $d_j - 1$ *largest singular values of* $\mathbf{Z}_j^*$ *are equal and have value larger than* $\frac{\text{tr}(\mathbf{\Pi}_j)}{d_j}$,

*where* $\mathbf{Z}_j^* \in \mathbb{R}^{d \times \text{tr}(\mathbf{\Pi}_j)}$ *denotes* $\mathbf{Z}^* \mathbf{\Pi}_j$ *with zero columns removed.*

## A.5 Proof of Main Results

We start with presenting a lemma that will be used in the proof to Theorem A.6.

**Lemma A.7.** *Given any twice differentiable* $f : \mathbb{R}_+ \to \mathbb{R}$, *integer* $r \in \mathbb{Z}_{++}$ *and* $c \in \mathbb{R}_+$, *consider the optimization problem*

$$\max_{\boldsymbol{x}} \sum_{p=1}^r f(x_p)$$

$$\text{s.t. } \boldsymbol{x} = [x_1, \ldots, x_r] \in \mathbb{R}_+^r, \ x_1 \geq x_2 \geq \cdots \geq x_r, \ \text{and} \ \sum_{p=1}^r x_p = c. \tag{16}$$

*Let $\boldsymbol{x}^*$ be an arbitrary global solution to* (16). *If the conditions*

- $f'(0) < f'(x)$ *for all* $x > 0$,

- *There exists* $x_T > 0$ *such that* $f'(x)$ *is strictly increasing in* $[0, x_T]$ *and strictly decreasing in* $[x_T, \infty)$,

- $f''(\frac{c}{r}) < 0$ *(equivalently, $\frac{c}{r} > x_T$)*,

*are satisfied, then we have either*

- $\boldsymbol{x}^* = [\frac{c}{r}, \ldots, \frac{c}{r}]$, *or*

- $\boldsymbol{x}^* = [x_H, \ldots, x_H, x_L]$ *for some* $x_H \in (\frac{c}{r}, \frac{c}{r-1})$ *and* $x_L > 0$.

*Proof.* The result holds trivially if $r = 1$. Throughout the proof we consider the case where $r > 1$.

We consider the optimization problem with the inequality constraint $x_1 \geq \cdots \geq x_r$ in (16) removed:

$$\max_{\boldsymbol{x}=[x_1,\ldots,x_r]\in\mathbb{R}_+^r} \sum_{p=1}^{r} f(x_p) \quad \text{s.t.} \sum_{p=1}^{r} x_p = c. \tag{17}$$

We need to show that any global solution $\boldsymbol{x}^* = [x_1^*, \ldots, x_r^*]$ to (17) is either $\boldsymbol{x}^* = [\frac{c}{r}, \ldots, \frac{c}{r}]$ or $\boldsymbol{x}^* = [x_H, \ldots, x_H, x_L] \cdot \boldsymbol{P}$ for some $x_H > \frac{c}{r}$, $x_L > 0$ and permutation matrix $\boldsymbol{P} \in \mathbb{R}^{r \times r}$. Let

$$\mathcal{L}(\boldsymbol{x}, \boldsymbol{\lambda}) = \sum_{p=1}^{r} f(x_p) - \lambda_0 \cdot \left( \sum_{p=1}^{r} x_p - c \right) - \sum_{p=1}^{r} \lambda_p x_p$$

be the Lagragian function for (17) where $\boldsymbol{\lambda} = [\lambda_0, \lambda_1, \ldots, \lambda_r]$ is the Lagragian multiplier. By the first order optimality conditions (i.e., the Karush–Kuhn–Tucker (KKT) conditions, see, e.g., [NW06, Theorem 12.1]), there exists $\boldsymbol{\lambda}^* = [\lambda_0^*, \lambda_1^*, \ldots, \lambda_r^*]$ such that

$$\sum_{p=1}^{r} x_q^* = c, \tag{18}$$

$$x_q^* \geq 0, \ \forall q \in \{1, \ldots, r\}, \tag{19}$$

$$\lambda_q^* \geq 0, \ \forall q \in \{1, \ldots, r\}, \tag{20}$$

$$\lambda_q^* \cdot x_q^* = 0, \ \forall q \in \{1, \ldots, r\}, \ \text{and} \tag{21}$$

$$[f'(x_1^*), \ldots, f'(x_r^*)] = [\lambda_0^*, \ldots, \lambda_0^*] + [\lambda_1^*, \ldots, \lambda_r^*]. \tag{22}$$

By using the KKT conditions, we first show that all entries of $\boldsymbol{x}^*$ are strictly positive. To prove by contradiction, suppose that $\boldsymbol{x}^*$ has $r_0$ nonzero entries and $r - r_0$ zero entries for some $1 \leq r_0 < r$. Note that $r_0 \geq 1$ since an all zero vector $\boldsymbol{x}^*$ does not satisfy the equality constraint (18).

Without loss of generality, we may assume that $x_p^* > 0$ for $p \leq r_0$ and $x_p^* = 0$ otherwise. By (21), we have

$$\lambda_1^* = \cdots = \lambda_{r_0}^* = 0.$$

Plugging it into (22), we get

$$f'(x_1^*) = \cdots = f'(x_{r_0}^*) = \lambda_0^*.$$

From (22) and noting that $x_{r_0+1} = 0$ we get

$$f'(0) = f'(x_{r_0+1}) = \lambda_0^* + \lambda_{r_0+1}^*.$$

Finally, from (20), we have

$$\lambda_{r_0+1}^* \geq 0.$$

Combining the last three equations above gives $f'(0) - f'(x_1^*) \geq 0$, contradicting the assumption that $f'(0) < f'(x)$ for all $x > 0$. This shows that $r_0 = r$, i.e., all entries of $\boldsymbol{x}^*$ are strictly positive. Using this fact and (21) gives

$$\lambda_p^* = 0 \ \text{ for all } p \in \{1, \ldots, r\}.$$

Combining this with (22) gives

$$f'(x_1^*) = \cdots = f'(x_r^*) = \lambda_0^*. \tag{23}$$

It follows from the fact that $f'(x)$ is strictly unimodal that

$$\exists \, x_H \geq x_L > 0 \ \text{s.t.} \ \{x_p^*\}_{p=1}^{r} \subseteq \{x_L, x_H\}. \tag{24}$$

That is, the set $\{x_p^*\}_{p=1}^{r}$ may contain no more than two values. To see why this is true, suppose that there exists three distinct values for $\{x_p^*\}_{p=1}^{r}$. Without loss of generality we may assume that $0 < x_1^* < x_2^* < x_3^*$. If $x_2^* \leq x_T$ (recall $x_T := \arg\max_{x \geq 0} f'(x)$), then by using the fact that $f'(x)$ is strictly increasing in $[0, x_T]$, we must have $f'(x_1^*) < f'(x_2^*)$ which contradicts (23). A similar contradiction is arrived by considering $f'(x_2^*)$ and $f'(x_3^*)$ for the case where $x_2^* > x_T$.

There are two possible cases as a consequence of (24). First, if $x_L = x_H$, then we have $x_1^* = \cdots = x_r^*$. By further using (18) we get

$$x_1^* = \cdots = x_r^* = \frac{c}{r}.$$

It remains to consider the case where $x_L < x_H$. First, by the unimodality of $f'(x)$, we must have $x_L < x_T < x_H$, therefore

$$f''(x_L) > 0 \text{ and } f''(x_H) < 0. \tag{25}$$

Let $\ell := |\{p : x_p = x_L\}|$ be the number of entries of $\boldsymbol{x}^*$ that are equal to $x_L$ and $h := r - \ell$. We show that it is necessary to have $\ell = 1$ and $h = r - 1$. To prove by contradiction, assume that $\ell > 1$ and $h < r - 1$. Without loss of generality we may assume $\{x_p^* = x_H\}_{p=1}^h$ and $\{x_p^* = x_L\}_{p=h+1}^r$. By (25), we have

$$f''(x_p^*) > 0 \text{ for all } p > h.$$

In particular, by using $h < r - 1$ we have

$$f''(x_{r-1}^*) > 0 \text{ and } f''(x_r^*) > 0. \tag{26}$$

On the other hand, by using the second order necessary conditions for constraint optimization (see, e.g., [NW06, Theorem 12.5]), the following result holds

$$\boldsymbol{v}^\top \nabla_{\boldsymbol{xx}} \mathcal{L}(\boldsymbol{x}^*, \boldsymbol{\lambda}^*) \boldsymbol{v} \le 0, \text{ for all } \left\{ \boldsymbol{v} : \left\langle \nabla_{\boldsymbol{x}} \left( \sum_{p=1}^r x_p^* - c \right), \boldsymbol{v} \right\rangle = 0 \right\}$$

$$\iff \sum_{p=1}^r f''(x_p^*) \cdot v_p^2 \le 0, \text{ for all } \left\{ \boldsymbol{v} = [v_1, \ldots, v_r] : \sum_{p=1}^r v_p = 0 \right\}. \tag{27}$$

Take $\boldsymbol{v}$ to be such that $v_1 = \cdots = v_{r-2} = 0$ and $v_{r-1} = -v_r \ne 0$. Plugging it into (27) gives

$$f''(x_{r-1}^*) + f''(x_r^*) \le 0,$$

which contradicts (26). Therefore, we may conclude that $\ell = 1$. That is, $\boldsymbol{x}^*$ is given by

$$\boldsymbol{x}^* = [x_H, \ldots, x_H, x_L], \text{ where } x_H > x_L > 0.$$

By using the condition in (18), we may further show that

$$(r-1)x_H + x_L = c \implies x_H = \frac{c}{r-1} - \frac{c}{x_L} < \frac{x_L}{r-1},$$

$$(r-1)x_H + x_L = c \implies (r-1)x_H + x_H > c \implies x_H > \frac{c}{r},$$

which completes our proof. $\qquad\square$

*Proof of Theorem A.6.* Without loss of generality, let $\boldsymbol{Z}^* = [\boldsymbol{Z}_1^*, \ldots, \boldsymbol{Z}_k^*]$ be the optimal solution of problem (15).

To show that $\boldsymbol{Z}_j^*, j \in \{1, \ldots, k\}$ are pairwise orthogonal, suppose for the purpose of arriving at a contradiction that $(\boldsymbol{Z}_{j_1}^*)^\top \boldsymbol{Z}_{j_2}^* \ne \boldsymbol{0}$ for some $1 \le j_1 < j_2 \le k$. By using Lemma A.5, the strict inequality in (14) holds for the optimal solution $\boldsymbol{Z}^*$. That is,

$$\Delta R(\boldsymbol{Z}^*, \boldsymbol{\Pi}, \epsilon) < \sum_{j=1}^k \frac{1}{2m} \log \left( \frac{\det^m \left( \boldsymbol{I} + \frac{d}{m\epsilon^2} \boldsymbol{Z}_j^* (\boldsymbol{Z}_j^*)^\top \right)}{\det^{m_j} \left( \boldsymbol{I} + \frac{d}{m_j \epsilon^2} \boldsymbol{Z}_j^* (\boldsymbol{Z}_j^*)^\top \right)} \right). \tag{28}$$

On the other hand, since $\sum_{j=1}^k d_j \le d$, there exists $\{\boldsymbol{U}_j' \in \mathbb{R}^{d \times d_j}\}_{j=1}^k$ such that the columns of the matrix $[\boldsymbol{U}_1', \ldots, \boldsymbol{U}_k']$ are orthonormal. Denote $\boldsymbol{Z}_j^* = \boldsymbol{U}_j^* \boldsymbol{\Sigma}_j^* (\boldsymbol{V}_j^*)^\top$ the compact SVD of $\boldsymbol{Z}_j^*$, and let

$$\boldsymbol{Z}' = [\boldsymbol{Z}_1', \ldots, \boldsymbol{Z}_k'], \text{ where } \boldsymbol{Z}_j' = \boldsymbol{U}_j' \boldsymbol{\Sigma}_j^* (\boldsymbol{V}_j^*)^\top.$$

It follows that

$$(\boldsymbol{Z}_{j_1}')^\top \boldsymbol{Z}_{j_2}' = \boldsymbol{V}_{j_1}^* \boldsymbol{\Sigma}_{j_1}^* (\boldsymbol{U}_{j_1}')^\top \boldsymbol{U}_{j_2}' \boldsymbol{\Sigma}_{j_2}^* (\boldsymbol{V}_{j_2}^*)^\top = \boldsymbol{V}_{j_1}^* \boldsymbol{\Sigma}_{j_1}^* \boldsymbol{0} \boldsymbol{\Sigma}_{j_2}^* (\boldsymbol{V}_{j_2}^*)^\top = \boldsymbol{0} \text{ for all } 1 \le j_1 < j_2 \le k.$$

That is, the matrices $\boldsymbol{Z}_1', \ldots, \boldsymbol{Z}_k'$ are pairwise orthogonal. Applying Lemma A.5 for $\boldsymbol{Z}'$ gives

$$\Delta R(\boldsymbol{Z}', \boldsymbol{\Pi}, \epsilon) = \sum_{j=1}^k \frac{1}{2m} \log \left( \frac{\det^m \left( \boldsymbol{I} + \frac{d}{m\epsilon^2} \boldsymbol{Z}_j' (\boldsymbol{Z}_j')^\top \right)}{\det^{m_j} \left( \boldsymbol{I} + \frac{d}{m_j \epsilon^2} \boldsymbol{Z}_j' (\boldsymbol{Z}_j')^\top \right)} \right)$$

$$= \sum_{j=1}^k \frac{1}{2m} \log \left( \frac{\det^m \left( \boldsymbol{I} + \frac{d}{m\epsilon^2} \boldsymbol{Z}_j^* (\boldsymbol{Z}_j^*)^\top \right)}{\det^{m_j} \left( \boldsymbol{I} + \frac{d}{m_j \epsilon^2} \boldsymbol{Z}_j^* (\boldsymbol{Z}_j^*)^\top \right)} \right), \tag{29}$$

where the second equality follows from Lemma A.3. Comparing (28) and (29) gives $\Delta R(\mathbf{Z}', \mathbf{\Pi}, \epsilon) > \Delta R(\mathbf{Z}^*, \mathbf{\Pi}, \epsilon)$, which contradicts the optimality of $\mathbf{Z}^*$. Therefore, we must have

$$(\mathbf{Z}_{j_1}^*)^\top \mathbf{Z}_{j_2}^* = \mathbf{0} \text{ for all } 1 \le j_1 < j_2 \le k.$$

Moreover, from Lemma A.3 we have

$$\Delta R(\mathbf{Z}^*, \mathbf{\Pi}, \epsilon) = \sum_{j=1}^k \frac{1}{2m} \log \left( \frac{\det^m \left( \mathbf{I} + \frac{d}{m\epsilon^2} \mathbf{Z}_j^*(\mathbf{Z}_j^*)^\top \right)}{\det^{m_j} \left( \mathbf{I} + \frac{d}{m_j\epsilon^2} \mathbf{Z}_j^*(\mathbf{Z}_j^*)^\top \right)} \right). \tag{30}$$

We now prove the result concerning the singular values of $\mathbf{Z}_j^*$. To start with, we claim that the following result holds:

$$\mathbf{Z}_j^* \in \arg\max_{\mathbf{Z}_j} \ \log \left( \frac{\det^m \left( \mathbf{I} + \frac{d}{m\epsilon^2} \mathbf{Z}_j \mathbf{Z}_j^\top \right)}{\det^{m_j} \left( \mathbf{I} + \frac{d}{m_j\epsilon^2} \mathbf{Z}_j \mathbf{Z}_j^\top \right)} \right) \quad \text{s.t. } \|\mathbf{Z}_j\|_F^2 = m_j, \ \mathsf{rank}(\mathbf{Z}_j) \le d_j. \tag{31}$$

To see why (31) holds, suppose that there exists $\widetilde{\mathbf{Z}}_j$ such that $\|\widetilde{\mathbf{Z}}_j\|_F^2 = m_j$, $\mathsf{rank}(\widetilde{\mathbf{Z}}_j) \le d_j$ and

$$\log \left( \frac{\det^m \left( \mathbf{I} + \frac{d}{m\epsilon^2} \widetilde{\mathbf{Z}}_j \widetilde{\mathbf{Z}}_j^\top \right)}{\det^{m_j} \left( \mathbf{I} + \frac{d}{m_j\epsilon^2} \widetilde{\mathbf{Z}}_j \widetilde{\mathbf{Z}}_j^\top \right)} \right) > \log \left( \frac{\det^m \left( \mathbf{I} + \frac{d}{m\epsilon^2} \mathbf{Z}_j^*(\mathbf{Z}_j^*)^\top \right)}{\det^{m_j} \left( \mathbf{I} + \frac{d}{m_j\epsilon^2} \mathbf{Z}_j^*(\mathbf{Z}_j^*)^\top \right)} \right). \tag{32}$$

Denote $\widetilde{\mathbf{Z}}_j = \widetilde{\mathbf{U}}_j \widetilde{\mathbf{\Sigma}}_j \widetilde{\mathbf{V}}_j^\top$ the compact SVD of $\widetilde{\mathbf{Z}}_j$ and let

$$\mathbf{Z}' = [\mathbf{Z}_1^*, \ldots, \mathbf{Z}_{j-1}^*, \mathbf{Z}_j', \mathbf{Z}_{j+1}^*, \ldots, \mathbf{Z}_k^*], \quad \text{where } \mathbf{Z}_j' := \mathbf{U}_j^* \widetilde{\mathbf{\Sigma}}_j \widetilde{\mathbf{V}}_j^\top.$$

Note that $\|\mathbf{Z}_j'\|_F^2 = m_j$, $\mathsf{rank}(\mathbf{Z}_j') \le d_j$ and $(\mathbf{Z}_j')^\top \mathbf{Z}_{j'}' = \mathbf{0}$ for all $j' \ne j$. It follows that $\mathbf{Z}'$ is a feasible solution to (15) and that the components of $\mathbf{Z}'$ are pairwise orthogonal. By using Lemma A.5, Lemma A.3 and (32) we have

$$\Delta R(\mathbf{Z}', \mathbf{\Pi}, \epsilon)$$

$$= \frac{1}{2m} \log \left( \frac{\det^m \left( \mathbf{I} + \frac{d}{m\epsilon^2} \mathbf{Z}_j'(\mathbf{Z}_j')^\top \right)}{\det^{m_j} \left( \mathbf{I} + \frac{d}{m_j\epsilon^2} \mathbf{Z}_j'(\mathbf{Z}_j')^\top \right)} \right) + \sum_{j' \ne j} \frac{1}{2m} \log \left( \frac{\det^m \left( \mathbf{I} + \frac{d}{m\epsilon^2} \mathbf{Z}_{j'}^*(\mathbf{Z}_{j'}^*)^\top \right)}{\det^{m_{j'}} \left( \mathbf{I} + \frac{d}{m_{j'}\epsilon^2} \mathbf{Z}_{j'}^*(\mathbf{Z}_{j'}^*)^\top \right)} \right)$$

$$= \frac{1}{2m} \log \left( \frac{\det^m \left( \mathbf{I} + \frac{d}{m\epsilon^2} \widetilde{\mathbf{Z}}_j(\widetilde{\mathbf{Z}}_j)^\top \right)}{\det^{m_j} \left( \mathbf{I} + \frac{d}{m_j\epsilon^2} \widetilde{\mathbf{Z}}_j(\widetilde{\mathbf{Z}}_j)^\top \right)} \right) + \sum_{j' \ne j} \frac{1}{2m} \log \left( \frac{\det^m \left( \mathbf{I} + \frac{d}{m\epsilon^2} \mathbf{Z}_{j'}^*(\mathbf{Z}_{j'}^*)^\top \right)}{\det^{m_{j'}} \left( \mathbf{I} + \frac{d}{m_{j'}\epsilon^2} \mathbf{Z}_{j'}^*(\mathbf{Z}_{j'}^*)^\top \right)} \right)$$

$$> \frac{1}{2m} \log \left( \frac{\det^m \left( \mathbf{I} + \frac{d}{m\epsilon^2} \mathbf{Z}_j^*(\mathbf{Z}_j^*)^\top \right)}{\det^{m_j} \left( \mathbf{I} + \frac{d}{m_j\epsilon^2} \mathbf{Z}_j^*(\mathbf{Z}_j^*)^\top \right)} \right) + \sum_{j' \ne j} \frac{1}{2m} \log \left( \frac{\det^m \left( \mathbf{I} + \frac{d}{m\epsilon^2} \mathbf{Z}_{j'}^*(\mathbf{Z}_{j'}^*)^\top \right)}{\det^{m_{j'}} \left( \mathbf{I} + \frac{d}{m_{j'}\epsilon^2} \mathbf{Z}_{j'}^*(\mathbf{Z}_{j'}^*)^\top \right)} \right)$$

$$= \sum_{j=1}^k \frac{1}{2m} \log \left( \frac{\det^m \left( \mathbf{I} + \frac{d}{m\epsilon^2} \mathbf{Z}_j^*(\mathbf{Z}_j^*)^\top \right)}{\det^{m_j} \left( \mathbf{I} + \frac{d}{m_j\epsilon^2} \mathbf{Z}_j^*(\mathbf{Z}_j^*)^\top \right)} \right).$$

Combining it with (30) shows $\Delta R(\mathbf{Z}', \mathbf{\Pi}, \epsilon) > \Delta R(\mathbf{Z}^*, \mathbf{\Pi}, \epsilon)$, contradicting the optimality of $\mathbf{Z}^*$. Therefore, the result in (31) holds.

Observe that the optimization problem in (31) depends on $\mathbf{Z}_j$ only through its singular values. That is, by letting $\boldsymbol{\sigma}_j := [\sigma_{1,j}, \ldots, \sigma_{\min(m_j,d),j}]$ be the singular values of $\mathbf{Z}_j$, we have

$$\log \left( \frac{\det^m \left( \mathbf{I} + \frac{d}{m\epsilon^2} \mathbf{Z}_j \mathbf{Z}_j^\top \right)}{\det^{m_j} \left( \mathbf{I} + \frac{d}{m_j\epsilon^2} \mathbf{Z}_j \mathbf{Z}_j^\top \right)} \right) = \sum_{p=1}^{\min\{m_j,d\}} \log \left( \frac{(1 + \frac{d}{m\epsilon^2} \sigma_{p,j}^2)^m}{(1 + \frac{d}{m_j\epsilon^2} \sigma_{p,j}^2)^{m_j}} \right),$$

also, we have

$$\|\mathbf{Z}_j\|_F^2 = \sum_{p=1}^{\min\{m_j,d\}} \sigma_{p,j}^2 \text{ and } \mathsf{rank}(\mathbf{Z}_j) = \|\boldsymbol{\sigma}_j\|_0.$$

Using these relations, (31) is equivalent to

$$
\max_{\boldsymbol{\sigma}_j \in \mathbb{R}_+^{\min\{m_j,d\}}} \sum_{p=1}^{\min\{m_j,d\}} \log \left( \frac{(1 + \frac{d}{m\epsilon^2}\sigma_{p,j}^2)^m}{(1 + \frac{d}{m_j\epsilon^2}\sigma_{p,j}^2)^{m_j}} \right)
$$

$$
\text{s.t.} \sum_{p=1}^{\min\{m_j,d\}} \sigma_{p,j}^2 = m_j, \text{ and } \mathsf{rank}(\boldsymbol{Z}_j) = \|\boldsymbol{\sigma}_j\|_0 \tag{33}
$$

Let $\boldsymbol{\sigma}_j^* = [\sigma_{1,j}^*, \ldots, \sigma_{\min\{m_j,d\},j}^*]$ be an optimal solution to (33). Without loss of generality we assume that the entries of $\boldsymbol{\sigma}_j^*$ are sorted in descending order. It follows that

$$
\sigma_{p,j}^* = 0 \text{ for all } p > d_j,
$$

and

$$
[\sigma_{1,j}^*, \ldots, \sigma_{d_j,j}^*] = \underset{\substack{[\sigma_{1,j},\ldots,\sigma_{d_j,j}] \in \mathbb{R}_+^{d_j} \\ \sigma_{1,j} \geq \cdots \geq \sigma_{d_j,j}}}{\arg\max} \sum_{p=1}^{d_j} \log \left( \frac{(1 + \frac{d}{m\epsilon^2}\sigma_{p,j}^2)^m}{(1 + \frac{d}{m_j\epsilon^2}\sigma_{p,j}^2)^{m_j}} \right) \quad \text{s.t.} \sum_{p=1}^{d_j} \sigma_{p,j}^2 = m_j. \tag{34}
$$

Then we define

$$
f(x; d, \epsilon, m_j, m) = \log \left( \frac{(1 + \frac{d}{m\epsilon^2}x)^m}{(1 + \frac{d}{m_j\epsilon^2}x)^{m_j}} \right),
$$

and rewrite (34) as

$$
\max_{\substack{[x_1,\ldots,x_{d_j}] \in \mathbb{R}_+^{d_j} \\ x_1 \geq \cdots \geq x_{d_j}}} \sum_{p=1}^{d_j} f(x_p; d, \epsilon, m_j, m) \quad \text{s.t.} \sum_{p=1}^{d_j} x_p = m_j. \tag{35}
$$

We compute the first and second derivative for $f$ with respect to $x$, which are given by

$$
f'(x; d, \epsilon, m_j, m) = \frac{d^2 x (m - m_j)}{(dx + m\epsilon^2)(dx + m_j\epsilon^2)},
$$

$$
f''(x; d, \epsilon, m_j, m) = \frac{d^2(m - m_j)(mm_j\epsilon^4 - d^2x^2)}{(dx + m\epsilon^2)^2(dx + m_j\epsilon^2)^2}.
$$

Note that

- $0 = f'(0) < f'(x)$ for all $x > 0$,

- $f'(x)$ is strictly increasing in $[0, x_T]$ and strictly decreasing in $[x_T, \infty)$, where $x_T = \epsilon^2 \sqrt{\frac{m}{d}\frac{m_j}{d}}$, and

- by using the condition $\epsilon^4 < \frac{m_j}{m}\frac{d^2}{d_j^2}$, we have $f''(\frac{m_j}{d_j}) < 0$.

Therefore, we may apply Lemma A.7 and conclude that the unique optimal solution to (35) is either

- $\boldsymbol{x}^* = [\frac{m_j}{d_j}, \ldots, \frac{m_j}{d_j}]$, or

- $\boldsymbol{x}^* = [x_H, \ldots, x_H, x_L]$ for some $x_H \in (\frac{m_j}{d_j}, \frac{m_j}{d_j-1})$ and $x_L > 0$.

Equivalently, we have either

- $[\sigma_{1,j}^*, \ldots, \sigma_{d_j,j}^*] = \left[ \sqrt{\frac{m_j}{d_j}}, \ldots, \sqrt{\frac{m_j}{d_j}} \right]$, or

- $[\sigma_{1,j}^*, \ldots, \sigma_{d_j,j}^*] = [\sigma_H, \ldots, \sigma_H, \sigma_L]$ for some $\sigma_H \in \left( \sqrt{\frac{m_j}{d_j}}, \sqrt{\frac{m_j}{d_j-1}} \right)$ and $\sigma_L > 0$,

as claimed. $\qquad \square$

# B  Additional Simulations and Experiments

## B.1  Simulations - Verifying Diversity Promoting Properties of MCR$^2$

As proved in Theorem A.6, the proposed MCR$^2$ objective promotes within-class diversity. In this section, we use simulated data to verify the diversity promoting property of MCR$^2$. As shown in Table 3, we calculate our proposed MCR$^2$ objective on simulated data. We observe that orthogonal subspaces with *higher* dimension achieve higher MCR$^2$ value, which is consistent with our theoretical analysis in Theorem A.6.

Table 3: **MCR$^2$ objective on simulated data.** We evaluate the proposed MCR$^2$ objective defined in (8), including $R$, $R^c$, and $\Delta R$, on simulated data. The output dimension $d$ is set as 512, 256, and 128. We set the batch size as $m = 1000$ and random assign the label of each sample from 0 to 9, i.e., 10 classes. We generate two types of data: 1) (RANDOM GAUSSIAN) For comparison with data without structures, for each class we generate random vectors sampled from Gaussian distribution (the dimension is set as the output dimension $d$) and normalize each vector to be on the unit sphere. 2) (SUBSPACE) For each class, we generate vectors sampled from its corresponding subspace with dimension $d_j$ and normalize each vector to be on the unit sphere. We consider the subspaces from different classes are orthogonal/nonorthogonal to each other.

|  | $R$ | $R^c$ | $\Delta R$ | ORTHOGONAL? | OUTPUT DIMENSION |
|---|---|---|---|---|---|
| RANDOM GAUSSIAN | 552.70 | 193.29 | 360.41 | ✓ | 512 |
| SUBSPACE ($d_j = 50$) | 545.63 | 108.46 | **437.17** | ✓ | 512 |
| SUBSPACE ($d_j = 40$) | 487.07 | 92.71 | 394.36 | ✓ | 512 |
| SUBSPACE ($d_j = 30$) | 413.08 | 74.84 | 338.24 | ✓ | 512 |
| SUBSPACE ($d_j = 20$) | 318.52 | 54.48 | 264.04 | ✓ | 512 |
| SUBSPACE ($d_j = 10$) | 195.46 | 30.97 | 164.49 | ✓ | 512 |
| SUBSPACE ($d_j = 1$) | 31.18 | 4.27 | 26.91 | ✓ | 512 |
| RANDOM GAUSSIAN | 292.71 | 154.13 | 138.57 | ✓ | 256 |
| SUBSPACE ($d_j = 25$) | 288.65 | 56.34 | **232.31** | ✓ | 256 |
| SUBSPACE ($d_j = 20$) | 253.51 | 47.58 | 205.92 | ✓ | 256 |
| SUBSPACE ($d_j = 15$) | 211.97 | 38.04 | 173.93 | ✓ | 256 |
| SUBSPACE ($d_j = 10$) | 161.87 | 27.52 | 134.35 | ✓ | 256 |
| SUBSPACE ($d_j = 5$) | 98.35 | 15.55 | 82.79 | ✓ | 256 |
| SUBSPACE ($d_j = 1$) | 27.73 | 3.92 | 23.80 | ✓ | 256 |
| RANDOM GAUSSIAN | 150.05 | 110.85 | 39.19 | ✓ | 128 |
| SUBSPACE ($d_j = 12$) | 144.36 | 27.72 | **116.63** | ✓ | 128 |
| SUBSPACE ($d_j = 10$) | 129.12 | 24.06 | 105.05 | ✓ | 128 |
| SUBSPACE ($d_j = 8$) | 112.01 | 20.18 | 91.83 | ✓ | 128 |
| SUBSPACE ($d_j = 6$) | 92.55 | 16.04 | 76.51 | ✓ | 128 |
| SUBSPACE ($d_j = 4$) | 69.57 | 11.51 | 58.06 | ✓ | 128 |
| SUBSPACE ($d_j = 2$) | 41.68 | 6.45 | 35.23 | ✓ | 128 |
| SUBSPACE ($d_j = 1$) | 24.28 | 3.57 | 20.70 | ✓ | 128 |
| SUBSPACE ($d_j = 50$) | 145.60 | 75.31 | 70.29 | ✗ | 128 |
| SUBSPACE ($d_j = 40$) | 142.69 | 65.68 | 77.01 | ✗ | 128 |
| SUBSPACE ($d_j = 30$) | 135.42 | 54.27 | 81.15 | ✗ | 128 |
| SUBSPACE ($d_j = 20$) | 120.98 | 40.71 | 80.27 | ✗ | 128 |
| SUBSPACE ($d_j = 15$) | 111.10 | 32.89 | 78.21 | ✗ | 128 |
| SUBSPACE ($d_j = 12$) | 101.94 | 27.73 | 74.21 | ✗ | 128 |

## B.2  Implementation Details

**Training Setting.**  We mainly use ResNet-18 [HZRS16] in our experiments, where we use 4 residual blocks with layer widths $[64, 128, 256, 512]$. The implementation of network architectures used in this paper are mainly based on this github repo.[1] For data augmentation in the supervised setting, we apply the `RandomCrop` and `RandomHorizontalFlip`. For the supervised setting, we train the models for 500 epochs and use stage-wise learning rate decay every 200 epochs (decay by a factor of 10). For the supervised setting, we train the models for 100 epochs and use stage-wise learning rate decay at 20-th epoch and 40-th epoch (decay by a factor of 10).

**Evaluation Details.** For the supervised setting, we set the number of principal components for nearest subspace classifier $r_j = 30$. We also study the effect of $r_j$ in Section B.3.2. For the CIFAR100 dataset, we consider 20 superclasses and set the cluster number as 20, which is the same setting as in [CWM$^+$17, WXYL18].

**Datasets.** We apply the default datasets in PyTorch, including CIFAR10, CIFAR100, and STL10.

**Augmentations $\mathcal{T}$ used for the self-supervised setting.** We apply the same data augmentation for CIFAR10 dataset and CIFAR100 dataset and the pseudo-code is as follows.

```
import torchvision.transforms as transforms
TRANSFORM = transforms.Compose([
    transforms.RandomResizedCrop(32),
    transforms.RandomHorizontalFlip(),
    transforms.RandomApply([transforms.ColorJitter(0.4, 0.4, 0.4, 0.1)], p=0.8),
    transforms.RandomGrayscale(p=0.2),
    transforms.ToTensor()])
```

The augmentations we use for STL10 dataset and the pseudo-code is as follows.

```
import torchvision.transforms as transforms
TRANSFORM = transforms.Compose([
    transforms.RandomResizedCrop(96),
    transforms.RandomHorizontalFlip(),
    transforms.RandomApply([transforms.ColorJitter(0.8, 0.8, 0.8, 0.2)], p=0.8),
    transforms.RandomGrayscale(p=0.2),
    GaussianBlur(kernel_size=9),
    transforms.ToTensor()])
```

**Cross-entropy training details.** For CE models presented in Table 1, Figure 6(d)-6(f), and Figure 7, we use the same network architecture, ResNet-18 [HZRS16], for cross-entropy training on CIFAR10, and set the output dimension as 10 for the last layer. We apply SGD, and set learning rate `lr=0.1`, momentum `momentum=0.9`, and weight decay `wd= 5e-4`. We set the total number of training epoch as 400, and use stage-wise learning rate decay every 150 epochs (decay by a factor of 10).

## B.3 Additional Experimental Results

### B.3.1 PCA Results of MCR$^2$ Training versus Cross-Entropy Training

For comparison, similar to Figure 3(c), we calculate the principle components of representations learned by MCR$^2$ training and cross-entropy training. For cross-entropy training, we take the output of the second last layer as the learned representation. The results are summarized in Figure 6. We also compare the cosine similarity between learned representations for both MCR$^2$ training and cross-entropy training, and the results are presented in Figure 7.

As shown in Figure 6, we observe that representations learned by MCR$^2$ are much more diverse, the dimension of learned features (each class) is around a dozen, and the dimension of the overall features is nearly 120, and the output dimension is 128. In contrast, the dimension of the overall features learned using entropy is slightly greater than 10, which is much smaller than that learned by MCR$^2$. From Figure 7, for MCR$^2$ training, we find that the features of different class are almost orthogonal.

**Visualize representative images selected from CIFAR10 dataset by using MCR$^2$.** As mentioned in Section 1, obtaining the properties of desired representation in the proposed MCR$^2$ principle is equivalent to performing *nonlinear generalized principle components* on the given dataset. As shown in Figure 6(a)-6(c), MCR$^2$ can indeed learn such diverse and discriminative representations. In order to better interpret the representations learned by MCR$^2$, we select images according to their "principal" components (singular vectors using SVD) of the learned features. In Figure 8, we visualize images selected from class-'Bird' and class-'Ship'. For each class, we first compute top-10 singular

(a) PCA: MCR$^2$ training learned features for overall data (first 30 components).

(b) PCA: MCR$^2$ training learned features for overall data.

(c) PCA: MCR$^2$ training learned features for every class.

(d) PCA: cross-entropy training learned features for overall data (first 30 components).

(e) PCA: cross-entropy training learned features for overall data.

(f) PCA: cross-entropy training learned features for every class.

Figure 6: Principal component analysis (PCA) of learned representations for the MCR$^2$ trained model (**first row**) and the cross-entropy trained model (**second row**).

Figure 7: Cosine similarity between learned features by using the MCR$^2$ objective (**left**) and CE loss (**right**).

vectors of the SVD of the learned features and then for each of the top singular vectors, we display in each row the top-10 images whose corresponding features are closest to the singular vector. As shown in Figure 8, we observe that images in the same row share many common characteristics such as shapes, textures, patterns, and styles, whereas images in different rows are significantly different from each other – suggesting our method captures all the different "modes" of the data even within the same class. Notice that top rows are associated with components with larger singular values, hence they are images that show up more frequently in the dataset.

In Figure 9(a), we visualize the 10 "principal" images selected from CIFAR10 for each of the 10 classes. That is, for each class, we display the 10 images whose corresponding features are most coherent with the top-10 singular vectors. We observe that the selected images are much more diverse and representative than those selected randomly from the dataset (displayed on the CIFAR official website), indicating such principal images can be used as a good "summary" of the dataset.

(a) Bird         (b) Ship

Figure 8: Visualization of principal components learned for class 2-'Bird' and class 8-'Ship'. For each class $j$, we first compute the top-10 singular vectors of the SVD of the learned features $\boldsymbol{Z}_j$. Then for the $l$-th singular vector of class $j$, $\boldsymbol{u}_j^l$, and for the feature of the $i$-th image of class $j$, $\boldsymbol{z}_j^i$, we calculate the absolute value of inner product, $|\langle \boldsymbol{z}_j^i, \boldsymbol{u}_j^l \rangle|$, then we select the top-10 images according to $|\langle \boldsymbol{z}_j^i, \boldsymbol{u}_j^l \rangle|$ for each singular vector. In the above two figures, each row corresponds to one singular vector (component $C_l$). The rows are sorted based on the magnitude of the associated singular values, from large to small.

(a) 10 representative images from each class based on top-10 principal components of the SVD of learned representations by MCR$^2$.

(b) Randomly selected 10 images from each class.

Figure 9: Visualization of top-10 "principal" images for each class in the CIFAR10 dataset. **(a)** For each class-$j$, we first compute the top-10 singular vectors of the SVD of the learned features $\boldsymbol{Z}_j$. Then for the $l$-th singular vector of class $j$, $\boldsymbol{u}_j^l$, and for the feature of the $i$-th image of class $j$, $\boldsymbol{z}_j^i$, we calculate the absolute value of inner product, $|\langle \boldsymbol{z}_j^i, \boldsymbol{u}_j^l \rangle|$, then we select the largest one for each singular vector within class $j$. Each row corresponds to one class, and each image corresponds to one singular vector, ordered by the value of the associated singular value. **(b)** For each class, 10 images are randomly selected in the dataset. These images are the ones displayed in the CIFAR dataset website [Kri09].

### B.3.2    Experimental Results of MCR$^2$ in the Supervised Learning Setting.

**Training details for mainline experiment.** For the model presented in Figure 1 (**Right**) and Figure 3, we use ResNet-18 to parameterize $f(\cdot, \theta)$, and we set the output dimension $d = 128$, precision $\epsilon^2 = 0.5$, mini-batch size $m = 1,000$. We use SGD in Pytorch [PGM$^+$19] as the optimizer, and set the learning rate `lr=0.01`, weight decay `wd=5e-4`, and `momentum=0.9`.

**Experiments for studying the effect of hyperparameters and architectures.** We present the experimental results of MCR$^2$ training in the supervised setting by using various training hyperparameters and different network architectures. The results are summarized in Table 4. Besides the ResNet architecture, we also consider VGG architecture [SZ15] and ResNext achitecture [XGD$^+$17]. From Table 4, we find that larger batch size $m$ can lead to better performance. Also, models with higher output dimension $d$ require larger training batch size $m$.

Table 4: Experiments of MCR$^2$ in the supervised setting on the CIFAR10 dataset.

| ARCH | DIM $d$ | PRECISION $\epsilon^2$ | BATCHSIZE $m$ | LR | ACC | COMMENT |
|---|---|---|---|---|---|---|
| RESNET-18 | 128 | 0.5 | 1,000 | 0.01 | 92.20% | MAINLINE, FIG 3 |
| RESNEXT-29 | 128 | 0.5 | 1,000 | 0.01 | 92.55% | DIFFERENT |
| VGG-11 | 128 | 0.5 | 1,000 | 0.01 | 90.76% | ARCHITECTURE |
| RESNET-18 | 512 | 0.5 | 1,000 | 0.01 | 88.60% | EFFECT OF |
| RESNET-18 | 256 | 0.5 | 1,000 | 0.01 | 92.10% | OUTPUT |
| RESNET-18 | 64 | 0.5 | 1,000 | 0.01 | 92.21% | DIMENSION |
| RESNET-18 | 128 | 1.0 | 1,000 | 0.01 | 93.06% | EFFECT OF |
| RESNET-18 | 128 | 0.4 | 1,000 | 0.01 | 91.93% | PRECISION |
| RESNET-18 | 128 | 0.2 | 1,000 | 0.01 | 90.06% | |
| RESNET-18 | 128 | 0.5 | 500 | 0.01 | 82.33% | |
| RESNET-18 | 128 | 0.5 | 2,000 | 0.01 | 93.02% | |
| RESNET-18 | 128 | 0.5 | 4,000 | 0.01 | 92.59% | EFFECT OF |
| RESNET-18 | 512 | 0.5 | 2,000 | 0.01 | 92.47% | BATCH SIZE |
| RESNET-18 | 512 | 0.5 | 4,000 | 0.01 | 92.17% | |
| RESNET-18 | 128 | 0.5 | 1,000 | 0.05 | 86.02% | |
| RESNET-18 | 128 | 0.5 | 1,000 | 0.005 | 92.39% | EFFECT OF LR |
| RESNET-18 | 128 | 0.5 | 1,000 | 0.001 | 92.23% | |

**Effect of $r_j$ on classification.** Unless otherwise stated, we set the number of components $r_j = 30$ for nearest subspace classification. We study the effect of $r_j$ when used for classification, and the results are summarized in Table 5. We observe that the nearest subspace classification works for a wide range of $r_j$.

Table 5: Effect of number of components $r_j$ for nearest subspace classification in the supervised setting.

| NUMBER OF COMPONENTS | $r_j = 10$ | $r_j = 20$ | $r_j = 30$ | $r_j = 40$ | $r_j = 50$ |
|---|---|---|---|---|---|
| MAINLINE (LABEL NOISE RATIO=0.0) | 92.68% | 92.53% | 92.20% | 92.32% | 92.17% |
| LABEL NOISE RATIO=0.1 | 91.71% | 91.73% | 91.16% | 91.83% | 91.78% |
| LABEL NOISE RATIO=0.2 | 90.68% | 90.61% | 89.70% | 90.62% | 90.54% |
| LABEL NOISE RATIO=0.3 | 88.24% | 87.97% | 88.18% | 88.15% | 88.10% |
| LABEL NOISE RATIO=0.4 | 86.49% | 86.67% | 86.66% | 86.71% | 86.44% |
| LABEL NOISE RATIO=0.5 | 83.90% | 84.18% | 84.30% | 84.18% | 83.76% |

**Effect of $\epsilon^2$ on learning from corrupted labels.** To further study the proposed MCR$^2$ on learning from corrupted labels, we use different precision parameters, $\epsilon^2 = 0.75, 1.0$, in addition to the one shown in Table 1. Except for the precision parameter $\epsilon^2$, all the other parameters are the same as the mainline experiment (the first row in Table 4). The first row ($\epsilon^2 = 0.5$) in Table 6 is identical to the MCR$^2$ TRAINING in Table 2. Notice that with slightly different choices in $\epsilon^2$, one might even see slightly improved performance over the ones reported in the main body.

Table 6: Effect of Precision $\epsilon^2$ on classification results with features learned with labels corrupted at different levels by using MCR$^2$ training.

| PRECISION | RATIO=0.1 | RATIO=0.2 | RATIO=0.3 | RATIO=0.4 | RATIO=0.5 |
|---|---|---|---|---|---|
| $\epsilon^2 = 0.5$ | 91.16% | 89.70% | 88.18% | 86.66% | 84.30% |
| $\epsilon^2 = 0.75$ | **92.37%** | 90.82% | **89.91%** | **87.67%** | 83.69% |
| $\epsilon^2 = 1.0$ | 91.93% | **91.11%** | 89.60% | 87.09% | **84.53%** |

## B.4    Comparison with Related Work on Label Noise

We compare the proposed MCR$^2$ with OLE [LQMS18], Large Margin Deep Networks [EKM$^+$18], and ITLM [SS19] in label noise robustness experiments on CIFAR10 dataset. In Table 7, we compare MCR$^2$ with OLE [LQMS18] and Large Margin Deep Networks [EKM$^+$18] on the corrupted label task using the same network, MCR$^2$ achieves significant better performance. We compare MCR$^2$ with ITLM [SS19] using the same network. MCR2 achieves better performance without any noise ratio dependent hyperparameters as required by [SS19].

Table 7: Comparison with related work on learning from noisy labels.

| RESNET18 | RATIO=0.1 | RATIO=0.2 | RATIO=0.3 | RATIO=0.4 | RATIO=0.5 |
|---|---|---|---|---|---|
| OLE [LQMS18] | 91.04% | 86.01% | 80.69% | 71.79% | 61.06% |
| LARGEMARGIN [EKM$^+$18] | 90.10% | 87.42% | 83.77% | 78.51% | 72.48% |
| MCR$^2$ | **91.16%** | **89.70%** | **88.18%** | **86.66%** | **84.30%** |

| WRN16 | RATIO=0.1 | RATIO=0.3 | RATIO=0.5 | RATIO=0.7 | |
|---|---|---|---|---|---|
| ITLM [SS19] | 90.33% | 88.23% | 82.51% | 64.74% | |
| MCR$^2$ | **91.55%** | **88.81%** | **84.25%** | **67.09%** | |

### B.4.1    Experimental Results of MCR$^2$ in the Self-supervised Learning Setting

**Training details of MCR$^2$-CTRL.**    For three datasets (CIFAR10, CIFAR100, and STL10), we use ResNet-18 as in the supervised setting, and we set the output dimension $d = 128$, precision $\epsilon^2 = 0.5$, mini-batch size $k = 20$, number of augmentations $n = 50$, $\gamma_1 = \gamma_2 = 20$. We observe that MCR$^2$-CTRL can achieve better clustering performance by using smaller $\gamma_2$, i.e., $\gamma_2 = 15$, on CIFAR10 and CIFAR100 datasets. We use SGD in Pytorch [PGM$^+$19] as the optimizer, and set the learning rate `lr=0.1`, weight decay `wd=5e-4`, and `momentum=0.9`.

**Training dynamic comparison between MCR$^2$ and MCR$^2$-CTRL**    . In the self-supervised setting, we compare the training process for MCR$^2$ and MCR$^2$-CTRL in terms of $R, \widetilde{R}, R^c$, and $\Delta R$. For MCR$^2$ training, the features first expand (for both $R$ and $R^c$) then compress (for ). For MCR$^2$-CTRL, both $\widetilde{R}$ and $R^c$ first compress then $\widetilde{R}$ expands quickly and $R^c$ remains small, as we have seen in Figure 5 in the main body.

**Clustering results comparison.**    We compare the clustering performance between MCR$^2$ and MCR$^2$-CTRL in terms of NMI, ACC, and ARI. The clustering results are summarized in Table 8. We find that MCR$^2$-CTRL can achieve better performance for clustering.

Table 8: Clustering comparison between MCR$^2$ and MCR$^2$-CTRL on CIFAR10 dataset.

| | NMI | ACC | ARI |
|---|---|---|---|
| MCR$^2$ | 0.544 | 0.570 | 0.399 |
| MCR$^2$-CTRL | 0.630 | 0.684 | 0.508 |

### B.4.2    Clustering Metrics and More Results

We first introduce the definitions of normalized mutual information (NMI) [SG02], clustering accuracy (ACC), and adjusted rand index (ARI) [HA85].

**Normalized mutual information (NMI).** Suppose $Y$ is the ground truth partition and $C$ is the prediction partition. The NMI metric is defined as

$$\text{NMI}(Y, C) = \frac{\sum_{i=1}^{k} \sum_{j=1}^{s} |Y_i \cap C_j| \log\left(\frac{m|Y_i \cap C_j|}{|Y_i||C_j|}\right)}{\sqrt{\left(\sum_{i=1}^{k} |Y_i| \log\left(\frac{|Y_i|}{m}\right)\right)\left(\sum_{j=1}^{s} |C_j| \log\left(\frac{|C_j|}{m}\right)\right)}},$$

where $Y_i$ is the $i$-th cluster in $Y$ and $C_j$ is the $j$-th cluster in $C$, and $m$ is the total number of samples.

**Clustering accuracy (ACC).** Given $m$ samples, $\{(\boldsymbol{x}_i, \boldsymbol{y}_i)\}_{i=1}^m$. For the $i$-th sample $\boldsymbol{x}_i$, let $\boldsymbol{y}_i$ be its ground truth label, and let $\boldsymbol{c}_i$ be its cluster label. The ACC metric is defined as

$$\text{ACC}(\boldsymbol{Y}, \boldsymbol{C}) = \max_{\sigma \in S} \frac{\sum_{i=1}^m \mathbf{1}\{\boldsymbol{y}_i = \sigma(\boldsymbol{c}_i)\}}{m},$$

where $S$ is the set includes all the one-to-one mappings from cluster to label, and $\boldsymbol{Y} = [\boldsymbol{y}_1, \ldots, \boldsymbol{y}_m]$, $\boldsymbol{C} = [\boldsymbol{c}_1, \ldots, \boldsymbol{c}_m]$.

**Adjusted rand index (ARI).** Suppose there are $m$ samples, and let $Y$ and $C$ be two clustering of these samples, where $Y = \{Y_1, \ldots, Y_r\}$ and $C = \{C_1, \ldots, C_s\}$. Let $m_{ij}$ denote the number of the intersection between $Y_i$ and $C_j$, i.e., $m_{ij} = |Y_i \cap C_j|$. The ARI metric is defined as

$$\text{ARI} = \frac{\sum_{ij} \binom{m_{ij}}{2} - \left( \sum_i \binom{a_i}{2} \sum_j \binom{b_j}{2} \right) / \binom{m}{2}}{\frac{1}{2} \left( \sum_i \binom{a_i}{2} + \sum_j \binom{b_j}{2} \right) - \left( \sum_i \binom{a_i}{2} \sum_j \binom{b_j}{2} \right) / \binom{m}{2}},$$

where $a_i = \sum_j m_{ij}$ and $b_j = \sum_i m_{ij}$.

**Comparison with [JHV19, HMT+17].** We compare MCR$^2$ with IIC [JHV19] and IM-SAT [HMT+17] in Table 9. We find that MCR$^2$ outperforms IIC [JHV19] and IMSAT [HMT+17] on both CIFAR10 and CIFAR100 by a large margin. For STL10, [HMT+17] applied pretrained ImageNet models and [JHV19] used more data for training.

Table 9: Compare with [JHV19, HMT+17] on clustering.

| DATASET | METRIC | IIC | IMSAT | MCR$^2$-CTRL |
|---------|--------|-----|-------|--------------|
| CIFAR10 | NMI | - | - | **0.630** |
|         | ACC | 0.617 | 0.456 | **0.684** |
|         | ARI | - | - | **0.508** |
| CIFAR100 | NMI | - | - | **0.387** |
|          | ACC | 0.257 | 0.275 | **0.375** |
|          | ARI | - | - | **0.178** |

**More experiments on the effect of hyperparameters of MCR$^2$-CTRL.** We provide more experimental results of MCR$^2$-CTRL training in the self-supervised setting by varying training hyperparameters on the STL10 dataset. The results are summarized in Table 10. Notice that the choice of hyperparameters only has small effect on the performance with the MCR$^2$-CTRL objective. We may hypothesize that, in order to further improve the performance, one has to seek other, potentially better, control of optimization dynamics or strategies. We leave those for future investigation.

Table 10: Experiments of MCR$^2$-CTRL in the self-supervised setting on STL10 dataset.

| ARCH | PRECISION $\epsilon^2$ | LEARNING RATE LR | NMI | ACC | ARI |
|------|------------------------|------------------|-----|-----|-----|
| RESNET-18 | 0.5 | 0.1 | 0.446 | 0.491 | 0.290 |
| RESNET-18 | 0.75 | 0.1 | 0.450 | 0.484 | 0.288 |
| RESNET-18 | 0.25 | 0.1 | 0.447 | 0.489 | 0.293 |
| RESNET-18 | 0.5 | 0.2 | 0.477 | 0.473 | 0.295 |
| RESNET-18 | 0.5 | 0.05 | 0.444 | 0.496 | 0.293 |
| RESNET-18 | 0.25 | 0.05 | 0.454 | 0.489 | 0.294 |