[Reviews · NeurIPS 2020]

Review 1

Summary and Contributions: This paper proposes to learn features by increasing the inter-class incoherence. A basic assumption is that features for each class lie near a linear subspace. The method is justified by rate distortion argument. Experimental results demonstrate the effectiveness of such a design. After rebuttal, I'm seeing some of my concerns (e.g., comparison with OLE) being resolved. Some deeper part (e.g., connection with robustness and generalization) are not fully addressed, but I understand it is unfair to ask for one paper solve a whole big research topic. Given these, I keep my current rating.

Strengths: The paper justifies the max-incoherence design from an information theoretic perspective, rather than pure heuristics. Theorem 2.1 shows some nice properties of the optimal solution. The experiments on label corrupted data is interesting.

Weaknesses: Overall, the paper can be considered as applying existing objective/criterion in learning with subspaces into learning of deep features. As already mentioned in the paper, there are existing papers adopting a similar design. To this end, the authors should show more comparison against these relevant methods, e.g., OLE. It is not clear why a larger intra-class subspace dimension (compared with OLE) is helpful. Perhaps that means more intra-class structure being preserved, and improves robustness. And therefore, gains are observed for classification of label corrupted data. However, these connections are not crystal clear in the paper. In fact, a core problem in understanding deep learning is robustness and generalization. This paper sort of touches the problem, but does not really advance readers’ understanding on it.

Correctness: I didn't check proof details for Theorem 1. But intuitively they are correct. The empirical methodology is correct. In addition, I have a concern whether the rate distortion function relies on Gaussian assumption of data. The authors should provide more details on that in order to be self-contained.

Clarity: Yes

Relation to Prior Work: See weakness

Reproducibility: Yes

Additional Feedback: Does the OLE type loss have the same property as theorem 1. What is the connection between rank, nuclear norm and logdet? A theoretical analysis is highly appreciated.


Review 2

Summary and Contributions: The paper proposes a representation learning method based on the principle of maximum coding rate reduction. The objective has the effect of encouraging the rank of the covariance matrix of the learned representations to be large, while simultaneously encouraging the ranks of the covariances for each individual class to be small. Theoretical properties of minimizers of the proposed objective are derived. Experiments on image datasets demonstrate improved robustness to random label corruptions vs. standard cross-entropy training, and that representations trained in an unsupervised manner possess cluster structure corresponding to ground-truth labels.

Strengths: 1. The paper proposes an intuitively reasonable learning objective for representation learning. This work is relevant to the NeurIPS community given its longstanding interest in supervised and unsupervised representation learning using deep neural networks. 2. Some relevant properties of optimal solutions to the MCR^2 objective are shown theoretically. 3. The empirical results for the method are encouraging. In particular, it is notable that the MCR^2 objective gives additional robustness to random label corruptions without explicitly accounting for label noise in the objective. 4. A range of helpful ablation results are provided in the appendix.

Weaknesses: 1. In Section 2, the authors seek to contrast the MCR^2 objective against mutual information-based approaches like information bottleneck. My impression is that there is a close relationship between these methods that warrants further discussion in the text. After all, the MCR^2 objective admits the following interpretation: first fit Gaussian distributions to the z-vectors with Tikhonov regularization on the covariance matrices to account for possible degeneracy (i.e., the \eps parameter in the text), then compute the mutual information with class labels using the differential entropy of these (regularized) Gaussians. 2. The label noise robustness experiments can be improved by further comparing against other methods like iterative trimmed loss minimization [1]. 3. Some of the claims in the text are non-obvious and should be made with appropriate citations. For example, in L48 -- it is stated that the IB objective results in sacrificing "generalizability, robustness, or transferability". Such a statement does not appear to me to be obviously true. 4. The overall clarity of the presentation should be improved (see below). [1] Shen & Sanghavi. Learning with bad training data via iterative trimmed loss minimization. ICML 2019.

Correctness: The claims and method appear to be correct, although I did not check the proofs in detail.

Clarity: I found the ideas in the paper to be interesting, but the presentation was hindered by the inclusion of vague, hand-wavy prose. For example, the introduction states that the paper aims to learn "explicitly meaningful" representations for the data -- it is not clear what "meaningful" entails, or indeed how the learning objective described in the sequel relates to this stated goal. Additionally, there is an overabundance of parenthetical information and discussion in the form of footnotes. I found that this detracted from the flow of the narrative.

Relation to Prior Work: I was satisfied with the coverage of prior work in the paper, but the relationship with the information bottleneck method should be expanded on.

Reproducibility: Yes

Additional Feedback: After author response: Regarding the relation with information bottleneck: I concede that the MCR^2 objective doesn't seek a "minimal" representation in the sense that the mutual information between the input and representation is not penalized; however, it still appears that the method can be viewed as mutual information maximization between the representation and the labels, where the representation takes the form of a regularized Gaussian mixture. I would have preferred to see some more discussion on this point. Overall, I found that the submission presents an interesting idea with encouraging empirical results. While I would prefer a more focused presentation of the main ideas, I'm willing to increase my score by one.


Review 3

Summary and Contributions: - The paper proposes a compression-inspired learning framework that is applicable in supervised and unsupervised settings. - The authors present a theoretical analysis of the proposed framework showing that, for multi-class data, their loss assigns the embeddings of each class to subspaces that are orthogonal to all other subspaces. - The proposed framework is empirically evaluated on a variety of unsupervised and supervised tasks, on both synthetic and real data, and compared to prior work.

Strengths: - The paper gives a good overview over different recent representation learning techniques and discusses them under one umbrella. - The proposed learning framework is applicable in both the supervised setting, where it is shown to promote robustness, and in the unsupervised setting, where it achieves strong clustering results. - The paper aims to make progress towards learning of more structured, more interpretable representations, which is an important current research direction.

Weaknesses: - My main concern is that, I don’t see the benefits of modeling the data as a union of subspaces, where each subspace corresponds to a class, when the representation space is *learned*. In particular, since these subspaces won’t be orthogonal in practice, on real data. In an unsupervised setting, to recover the subspaces, one needs to perform subspace clustering, which is a hard problem and computationally expensive to perform. In a supervised setting, where estimation of the subspaces is easy, one needs to do nearest-subspace-classification which is more intricate than linear classification. In stark contrast, a linear head trained with a cross-entropy loss learns a representation space with approximately linearly separable regions for each class. As a consequence, classification is simple (linear) and Lp distances in representation space are meaningful (which is not necessarily the case when the classes lie on a union of subspaces). - I acknowledge the encouraging results regarding robustness of the representations learned with the proposed method. However, there are many other methods which can make neural networks with linear classification head more robust, for example [c]. Therefore I believe a union of subspace structure is not fundamentally required to achieve this. - While the theoretical analysis reveals interesting properties of the learned representation, it completely ignores the relationship between the individual data points and their representation, defined through the feature extractor. It is well-known that the structure and properties of the extractor crucially impact the learned representation, possibly even more than the loss, see e.g. [ZF14]. [c] Elsayed, Gamaleldin, et al. "Large margin deep networks for classification." Advances in neural information processing systems. 2018. --- Update after rebuttal: Thanks to the authors for their response. I now better see the benefits of encouraging orthogonality between class regions in the feature space, which is why I increased my rating. However, I'm still not sure whether the theoretical result is useful to explain what is going on, as I still believe the network architecture is crucial for the structure in the feature space. Furthermore, as pointed out by the other reviewers, the method seems to have many similarities with previous methods, which should be discussed more precisely.

Correctness: The experiments seem methodologically correct and are clearly described in the main paper and the appendix. The paper is accompanied by well-documented code to reproduce the results.

Clarity: The paper is well written and very polished. It is accompanied by a detailed appendix providing further details and proofs. Regarding Theorem 2.1. it should clearly be written that it applies to the supervised setting where the class assignments are given (which is stated in the main result in Theorem A.6). This is an important detail that should not be deferred to the appendix. It might make sense to revisit the footnotes. I feel that moving some of them into the main text might improve clarity.

Relation to Prior Work: While there is no dedicated related work section, related work is sufficiently covered. However, the following references also consider clustering-related representation learning and are evaluated mostly on the same data sets, with the same metrics, and should therefore be included and compared against. [a] Hu, Weihua, et al. "Learning Discrete Representations via Information Maximizing Self-Augmented Training." International Conference on Machine Learning. 2017. [b] Ji, Xu, João F. Henriques, and Andrea Vedaldi. "Invariant information clustering for unsupervised image classification and segmentation." Proceedings of the IEEE International Conference on Computer Vision. 2019.

Reproducibility: Yes

Additional Feedback: Minor comments: Computing the log det has cubic complexity and is known to be prone to instability. Did the authors encounter any problems along these lines? How is the simplex constraint on \Pi in (9) enforced in practice? Footnote 3: ...finitely *many* samples...


Review 4

Summary and Contributions: The authors propose maximal coding rate reduction as a means to learn diverse and discriminative low-dimensional representations from high-dimensional data. The authors empirically show that they establish the new state-of-the-art results in clustering mixed data and theoretically prove the theoretical guarantee for learning diverse features. They also demonstrate the robustness over cross-entropy based classification. Specifically, the authors argue that a good representation Z for data X is one that maximizes the coding rate reduction, defined by the difference between coding rate (derived from the rate-distortion function given the target distortion \e) for X itself and the coding rate for the mixed distribution, where the subset partition itself is either given or also optimized.

Strengths: The authors nicely bring information theoretic measures, such as the coding rate for the rate-distortion function, to propose the maximal coding rate reduction principle. The idea makes sense and sounds intuitive. They demonstrate pretty impressive performance gain on several datasets over existing baselines.

Weaknesses: Applying the MCR reduction to the large-scale dataset seems computationally very hard. Why MCR reduction results in robust features isn't unclear (as also noted by the authors). The gap is huge. It'll be interesting to study why this is the case.

Correctness: The claims and method look correct.

Clarity: The paper is well written and easy to follow and understand.

Relation to Prior Work: How this work differs from previous contributions is clearly discussed.

Reproducibility: No

Additional Feedback:

[Author Response · NeurIPS 2020]

**To All Reviewers:** We thank all reviews for your insightful feedback and your appreciation of our $MCR^2$ formulation.
We will incorporate suggestions on minor corrections, references, footnotes, and presentations in the final version.
**Why diverse intra-class representations?** This work aims to introduce a new objective (i.e., $MCR^2$) for learning
representations not only discriminative between classes as with cross-entropy loss, but also *diverse* within class. We
believe identifying more discriminative features lead to more reliable classification since the most discriminative
feature may not be present in all samples. We rigorously prove that this can be achieved with the proposed $MCR^2$ loss
function. Furthermore, we empirically demonstrate this objective can be used to train deep networks that have good
properties in handling label noise (in supervised setting) and achieve SOTA for clustering (in unsupervised setting).
**Robustness to label noise:** The initial motivation of $MCR^2$ is to
promote learning rich discriminative features. It is a nice surprise
that so learned deep features are more robust than existing learning
objectives including cross entropy and many others shown in Ta-
bles 1, 2. Unlike cross entropy that fits labels of individual samples,
$MCR^2$ compresses samples of each class *collectively*. As mentioned
in Section 4, given the compelling empirical evidence, a rigorous
justification of the robustness is an exciting problem for future work.

Table 1: Comparison with OLE and Large Margin [EKM+18] on learning from noisy labels.

| RESNET18 | RATIO=0.1 | RATIO=0.2 | RATIO=0.3 | RATIO=0.4 | RATIO=0.5 |
|---|---|---|---|---|---|
| OLE | 91.04% | 86.01% | 80.69% | 71.79% | 61.06% |
| [EKM+18] | 90.10% | 87.42% | 83.77% | 78.51% | 72.48% |
| $MCR^2$ | **91.16%** | **89.70%** | **88.18%** | **86.66%** | **84.30%** |

Table 2: Comparison with Trimmed Loss [SS19] on learning from noisy labels.

| WRN16 | RATIO=0.1 | RATIO=0.3 | RATIO=0.5 | RATIO=0.7 |
|---|---|---|---|---|
| [SS19] | 90.33% | 88.23% | 82.51% | 64.74% |
| $MCR^2$ | **91.55%** | **88.81%** | **84.25%** | **67.09%** |

**To Reviewer #1:** Please refer to the top of the rebuttal for the motivations of larger intra-class subspace in $MCR^2$.
**Q1:** *Compare with OLE: " 1). It is not clear why a larger . . . these connections are not crystal clear in the paper. 2).*
*Does the OLE type loss have the same property as Theorem 1? 3). The authors should show more comparison . . . OLE."*
**A1:** 1). We will make these connections more clear in the final version; 2). As mentioned in the paper (line 209-213),
OLE loss does *not* have the diversity property of $MCR^2$ given in Theorem 1; 3). In Table 1, we compare $MCR^2$ with
OLE on the corrupted label task using the same network architecture. $MCR^2$ achieves significantly better performance.
**Q2:** *Gaussian assumption of data: "I have a concern whether the rate distortion function . . . to be self-contained."*
**A2:** Thank you for your suggestion. As shown in [MDHW07], the rate distortion function can serve as a tight and
accurate approximation for a wide range of subspace-like distributions. We will give more details in the final version.
**Q3:** *The paper can be considered as applying existing objective/criterion . . . into learning of deep features.*
**A3:** We disagree. Our $MCR^2$ objective is new and is different from those in previous works such as OLE. To our best
knowledge, $MCR^2$ is *the first* objective theoretically shown to guarantee both diverse and discriminative properties.
**Q4:** *In fact, a core problem in understanding deep learning . . . on it.*
**A4:** Thanks for your comment. Precisely, we believe identifying a diverse and discriminative representation from the
data is an important step to gaining better understandings of the generalizability and robustness of deep learning.
**To Reviewer #2:**
**Q1:** *Relationship with information bottleneck (IB) framework: " In Section 2, the authors seek to . . . Gaussians."*
**A1:** Both $MCR^2$ and IB are information-theoretic objectives. However, the goal of IB is to find a *minimal* set of most
informative representations while $MCR^2$ aims to capture both diverse and discriminative representations, which is very
different. We will better clarify relationships with mutual information-based approaches in our final version.
**Q2:** *Related work on label noise: "The label noise robustness experiments . . . iterative trimmed loss minimization [1]."*
**A2:** In Table 2, we compare $MCR^2$ with [SS19] using the same network. $MCR^2$ achieves better performance *without*
any noise ratio dependent hyperparameters as required by [SS19]. We will add the comparison in the final version.
**To Reviewer #3:** Please refer to the top of the rebuttal for clarifying the objectives and motivations of our work.
**Q1:** *"My main concern is that, I don't see the benefits . . . lie on a union of subspaces)."*
**A1:** First of all, we do *not* model the original data by subspaces. $MCR^2$ can guide a deep network to map real data on
complicated nonlinear submanifolds to a union of orthogonal subspaces. Secondly, once the subspaces are learned, the
nearest subspace classification is computationally *efficient*. Finally, compared with *hidden* representations learned by
cross-entropy, the union of discriminative subspaces learned by $MCR^2$ is geometrically and statistically meaningful.
**Q2:** *"While the theoretical analysis reveals interesting properties . . . the loss, see e.g. [ZF14]."*
**A2:** Our theoretical analysis reveals that the proposed $MCR^2$ is optimized only when features are the most diverse and
discriminative. Our experiments have clearly shown that using $MCR^2$, deep features learned from real data such as
CIFAR10 have the same nice properties that are predicted by our theoretical results. We plan to rigorously justify this
phenomenon by studying the interplay of the $MCR^2$ objective and the choice of network architectures in future work.
**Q3:** *Related work on clustering: "While there is no dedicated related work . . . be included and compared against."*
**A3:** $MCR^2$ outperforms [HMT+17, JHV19] on both CIFAR10 and CIFAR100 by a large margin. For STL10, [HMT+17]
applied pretrained ImageNet models and $MCR^2$ outperforms [JHV19] when using the same amount of training data.
**Answers to minor comments:** We will add the above comparison and references, and also compared $MCR^2$ to
[EKM+18] on CIAFR10 with label noise (see Table 1). We did not encounter any computation issue when dealing with
$\log \det$ and the optimization is stable. The $\Pi$ is defined by the labels and satisfies the simplex constraint (footnote 15).
**To Reviewer #4:** Please refer to the top of the rebuttal for the question regarding the robustness of $MCR^2$.
**Q1:** *Applying the MCR reduction to the large-scale dataset seems computationally very hard.*
**A1:** The computation only increases *linearly* in the number of classes for the supervised learning setting.

[Meta-Review · NeurIPS 2020]

The paper proposed a method for learning representation that tries to maximize inter-class incoherence by embedding the data into orthogonal subspaces for different classes. While the reviewers recognize that this is an interesting idea, framed in a principle informational theoretic way, and the empirical results are promising, there are issues with clarity of the presentation and the connection to previous work. There is also a concern that the theory does not account for properties of the feature extractor, which are crucial for the properties of the embedding space. Given the promising empirical result, my recommendation is a weak accept.